# A CDK-regulated chromatin segregase promoting chromosome replication

Erika Chacin[1,8], Priyanka Bansal[1,8], Karl-Uwe Reusswig [2], Luis M. Diaz-Santin[3,4], Pedro Ortega[5], Petra Vizjak[1], Belen Gómez-González [5], Felix Müller-Planitz [1,6], Andrés Aguilera [5], Boris Pfander [2], Alan C. M. Cheung[3,4,7] & Christoph F. Kurat[1✉]

The replication of chromosomes during S phase is critical for cellular and organismal function. Replicative stress can result in genome instability, which is a major driver of cancer. Yet how chromatin is made accessible during eukaryotic DNA synthesis is poorly understood. Here, we report the characterization of a chromatin remodeling enzyme—Yta7—entirely distinct from classical SNF2-ATPase family remodelers. Yta7 is a AAA$^+$-ATPase that assembles into ~1 MDa hexameric complexes capable of segregating histones from DNA. The Yta7 chromatin segregase promotes chromosome replication both in vivo and in vitro. Biochemical reconstitution experiments using purified proteins revealed that the enzymatic activity of Yta7 is regulated by S phase-forms of Cyclin-Dependent Kinase (S-CDK). S-CDK phosphorylation stimulates ATP hydrolysis by Yta7, promoting nucleosome disassembly and chromatin replication. Our results present a mechanism for how cells orchestrate chromatin dynamics in co-ordination with the cell cycle machinery to promote genome duplication during S phase.

[1] Molecular Biology Division, Biomedical Center Munich, Ludwig-Maximilians-Universität, Munich, Planegg-Martinsried, Germany. [2] Max Planck Institute of Biochemistry, DNA Replication and Genome Integrity, Planegg-Martinsried, Germany. [3] Department of Structural and Molecular Biology, Institute of Structural and Molecular Biology, University College London, London, UK. [4] Institute of Structural and Molecular Biology, Biological Sciences, Birkbeck College, London, UK. [5] Andalusian Molecular Biology and Regenerative Medicine Centre-CABIMER, University of Seville-CSIC, Seville, Spain. [6] Present address: Institute of Physiological Chemistry, Medical Faculty Carl Gustav Carus, Technische Universität Dresden, Dresden, Germany. [7] Present address: School of Biochemistry, University of Bristol, Bristol, UK. [8] These authors contributed equally: Erika Chacin, Priyanka Bansal. ✉email: Christoph.kurat@bmc.med.lmu.de

The duplication of complex genomes is one of the most fundamental processes of life[1]. Eukaryotic genomes are tightly packaged into stable nucleosomes, the basic unit of chromatin[2], which were recently shown to represent a major barrier of eukaryotic DNA synthesis[3–5]. Eukaryotic replisomes alone cannot operate on chromatin efficiently and a minimal set of "chromatin factors", the ATP-dependent chromatin remodelers INO80 or ISW1a, the ATP-independent histone chaperone FACT/Nhp6, as well as SAGA and NuA4 histone acetyl transferases, are necessary for replication in vitro[5]. However, it is likely that other chromatin factors are required for cellular chromatin replication and its coordination with the cell cycle. Chromatin factors have vital roles beyond DNA replication, like transcription, DNA repair, or DNA damage signaling. It is currently unknown if mechanisms exist to distinguish chromatin factors to become S phase-specific to promote genome replication. Here, we identify the AAA$^+$-ATPase containing Yta7 protein from *Saccharomyces cerevisiae* as a novel factor promoting chromatin replication both in vivo and in vitro.

Yta7 was originally identified as a chromatin boundary element that influences histone gene transcription[6–8]. Interestingly, Yta7 is phosphorylated during the S phase by Cyclin-Dependent Kinase (CDK) and Casein Kinase 2 (CK2)[9]. Phosphorylation of Yta7 during the S phase caused its release from chromatin, allowing efficient histone gene transcription. One model at the time was that Yta7 might act as a boundary element to keep the histone chaperone Rtt106 and the chromatin remodeler RSC in place to prevent the spreading of a repressive chromatin structure into open reading frames of histone genes outside of the S phase. During the S phase, phosphorylation of Yta7 causes its eviction from chromatin, allowing efficient promoter escape and transcript elongation by RNA-Polymerase 2[9,10]. Another idea was that Yta7 might act as a chromatin-modifying enzyme, which uses ATP hydrolysis to actively manipulate chromatin structure[11,12]. Supporting this, *yta7* mutants exhibited increased nucleosome density[6,7,9,12–16]. However, to date, there is no biochemical data supporting either model. Furthermore, the role of S phase phosphorylation of Yta7 is still elusive.

Yta7 has roles beyond S phase histone gene transcription as well and is implicated in the assembly of nucleosomes at centromeric regions[17] and it was suggested that Abo1, Yta7′s homolog in *Schizosaccharomyces pombe*, might play a role in nucleosome assembly at transcribed regions[18,19].

Yta7 comprises a non-canonical bromodomain coupled to tandem AAA$^+$-ATPase domains where only one domain is well conserved and catalytically active, characteristic of the type-II family of AAA$^+$-ATPases[11,13,20]. Some AAA$^+$-ATPases are known to disrupt protein structures (unfoldases) or protein complexes (segregases). In particular, Cdc48/p97 segregase is known to recognize ubiquitylated proteins and to segregate these proteins from their endogenous protein complexes[21,22]. Cdc48 has been implicated in chromatin dynamics by manipulating chromatin-associated proteins like chromatin remodelers or RNA-polymerase II[23–25] but has not been described to directly modify chromatin structure.

Yta7 is conserved among eukaryotes with its human homolog ATAD2 emerging as an oncogene overexpressed in various cancers with poor prognosis[11,26–28]. Yet, the molecular functions of both Yta7 and ATAD2 remain obscure. Here we show that Yta7 serves as a chromatin segregase capable of disassembling nucleosomes to clear the major barrier of DNA synthesis. Remarkably, Yta7′s enzymatic function is regulated by CDK phosphorylation in the S phase, providing an explanation of how Yta7 chromatin segregase becomes S phase-specific to promote chromosome replication.

## Results

**Yta7 is involved in chromosome replication in vivo.** A genetic interaction, such as synthetic lethality, refers to an unexpected phenotype that arises from combining mutations in two or more genes and genes involved in the same biological process often show similar patterns or profiles of genetic interactions[29]. To shed light on *YTA7* function, we first explored *YTA7*′s genetic interaction profile within the global genetic interaction profile similarity network[30]. We found that besides genetic interaction profile similarity to genes with known roles in transcription, *YTA7* also exhibited genetic interactions in common with genes involved in nuclear transport, as well as in DNA replication and the response to replication stress (Fig. 1a, Supplementary Fig. 1a). Thus, in addition to its previously described function in transcription, *YTA7* may also have a role in DNA synthesis.

This is supported by the finding that Yta7 becomes phosphorylated during the S phase[9], and seven CDK phosphorylation sites cluster in close proximity to the canonical AAA$^+$-ATPase domain of Yta7 (Fig. 1b). The molecular consequences of these phosphorylation events, however, remain unknown. We now hypothesize that S phase phosphorylation by CDK might be linked to its ATPase function, possibly in a stimulating manner.

To address whether Yta7 is involved in chromosome replication, and to determine if its ATPase and phosphorylation by CDK are important for this, we used a K460A mutant of the Walker A motif within the active domain (*ATP-binding*) and a second mutant lacking the seven CDK phosphorylation sites threonine (T) 67, serine (S) 94, T 212, S 259, S 304, S 380, and T 445 as described previously[9] (*Phospho*) (Fig. 1b) and compared both mutants to a full *YTA7* deletion mutant (*Deletion*). All mutants were generated at the endogenous loci and expression levels were not affected (Supplementary Fig. 1b). To examine DNA replication in vivo, *YTA7* wild type and *yta7* mutant cells were arrested in the G1 phase with alpha-factor. After two quick washing steps, the cells were released into a fresh medium to resume cell cycle progression. The release from the alpha-factor block and S phase progression was monitored using fluorescence-activated cell sorting (FACS). Whilst all *yta7* mutants entered the S phase comparably to *YTA7* wild type (WT), progression through the S phase was delayed (Fig. 1c). This defect in S phase progression was similar for *ATP-binding*-mutants, as well as *Phospho*-mutants consistent with a role for Yta7 in this process, involving both its ATP binding activity and its phosphorylation by CDK. These results also support our idea that phosphorylation of Yta7 might function as a stimulatory signal for its ATPase activity. The ATP-binding activity and phosphorylation by CDK are critical functions of Yta7 during the S phase, as both *ATP-binding*-mutations and *Phospho*-mutations phenocopied the *Deletion*-mutant (Fig. 1c).

The observed replication defect was not due to activation of the DNA replication checkpoint or reduction of the S-CDK activity given that Rad53[31] and Orc6[32] phosphorylation were unaffected (Supplementary Fig. 1c), suggesting a different mechanism of impairment. As problems in S phase progression can lead to recombinogenic DNA damage, we measured Rad52 foci in S/G2 phase cells which defines centers of DNA repair and recombination[33]. Relative to WT, all *yta7* mutants showed a significant increase of Rad52 foci (Fig. 1d), which was accompanied by an increase in spontaneous levels of homologous recombination (Fig. 1e). These results suggest that Yta7 phosphorylation by CDK and its ATPase function prevent DNA damage during the S phase, lending support to the hypothesized role of Yta7 in maintaining replication fork progression. As mentioned above, Yta7 has a known role in S phase histone gene transcription. It is well established that imbalances of histone levels can cause genotoxic stress[34]. It is therefore possible that our in vivo phenotypes might be explained by a reduction of histone gene transcription caused by the

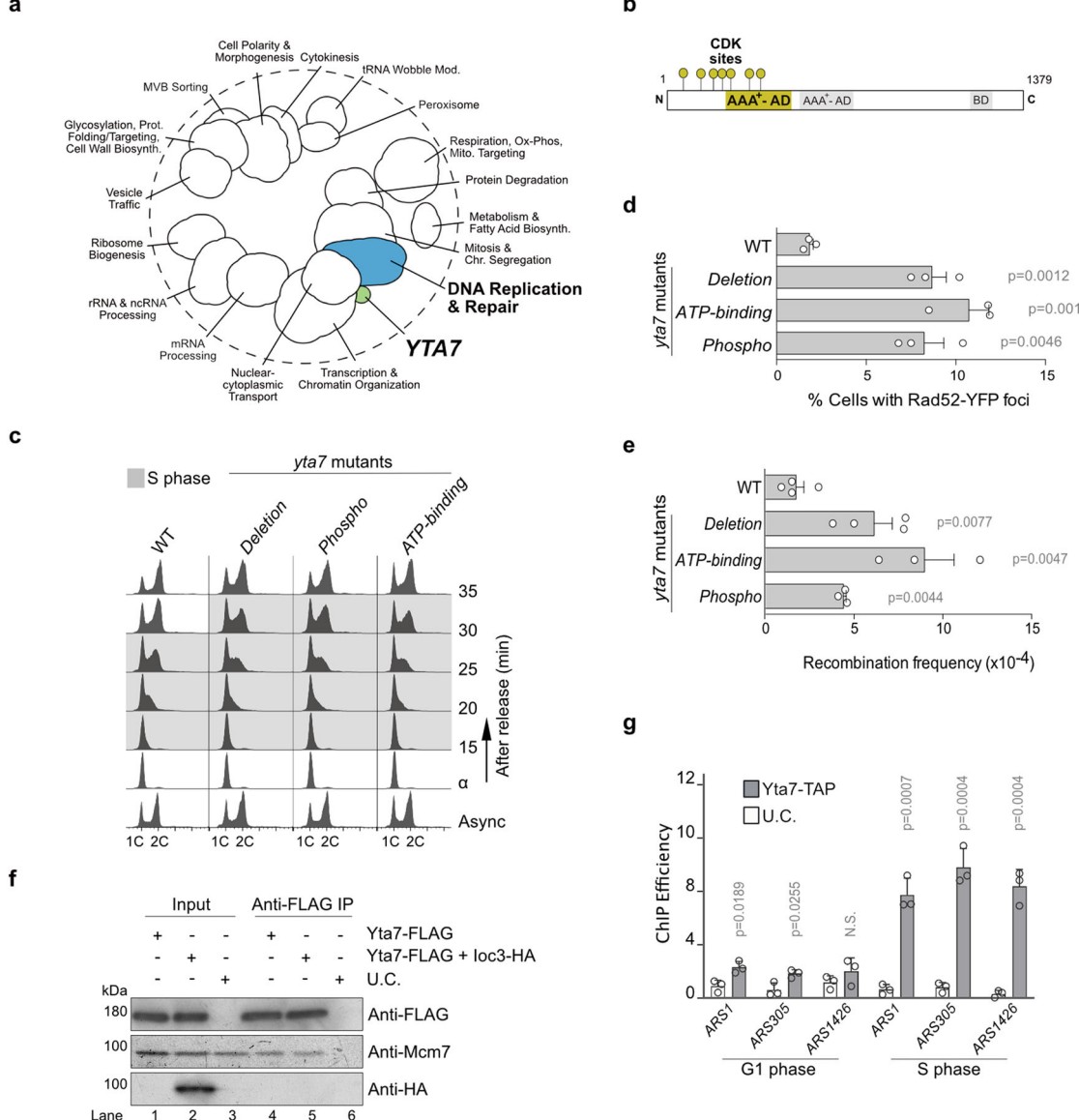

**Fig. 1 Yta7 has a role in chromosome replication in vivo. a** The position of *YTA7* in the global genetic interaction profile similarity network. **b** Domain organization of Yta7, highlighting the canonical AAA$^+$-ATPase domain and its adjacent CDK phosphorylation sites. AD: ATPase domain, BD: Bromodomain. **c** *YTA7* wild-type (WT), AAA$^+$-ATPase (*ATP-binding*), *yta7* CDK-phospho (*Phospho*), and *yta7Δ* (*Deletion*) mutant cells were synchronized in G1 phase with alpha-factor and released into S phase before monitoring DNA replication using FACS. Labels 1C and 2C indicate non-replicated or replicated DNA, respectively. FACS analyses have been replicated three times. **d** Accumulation of Rad52 foci and (**e**) increase of recombination frequency induced by *yta7* mutations compared to WT. **f** Co-immunoprecipitation experiment showing that Yta7 physically interacts with the replicative helicase Mcm2-7 but not with the chromatin remodeler ISW1a. Benzonase was added to prevent DNA-mediated interactions. Shown is a representative experiment, which has been biologically replicated three times. **g** Cells bearing Yta7-TAP were synchronized in the G1 phase using alpha-factor, released, and harvested during G1 and early S phases. Chromatin-Immunoprecipitation (ChIP) experiments were performed as described in "Methods" section. U.C. means untagged control. Mean values and standard deviations (S.D.) in **d, e,** and **g** were obtained from three biological replicates, except for WT and *Deletion* in **e**, which were repeated four times. *P*-values were obtained by using two-tailed unpaired *t*-test calculations. N.S. means statistically non-significant.

elimination of Yta7 function. In support of our hypothesis that Yta7 is important for chromosome replication and that our in vivo phenotypes are not the consequence of indirect effects, pull-down experiments plus benzonase treatment to inhibit DNA-mediated interactions showed that Yta7 physically associates with the replicative helicase Mcm2-7 (MCM) (Fig. 1f). As a negative control, Yta7 does not interact with the chromatin remodeling complex ISW1a (Fig. 1f). Furthermore, Chromatin Immunoprecipitation (ChIP) experiments revealed that Yta7 associates with a selection of early firing (*ARS1, ARS305,* and

*ARS1426*), as well as late firing origins of replication (*ARS501, ARS603,* and *ARS1412*) specifically during the S phase compared to the G1 phase (Fig. 1g and Supplementary Fig. 2).

Taken together, our in vivo analyses suggest that (i) Yta7 is important for replication and S phase progression and that (ii) CDK phosphorylation of Yta7 in the S phase might be important for ATPase function, which consequently supports chromosome replication. To prove these hypotheses we, therefore, set out to biochemically reconstitute Yta7 function with purified proteins.

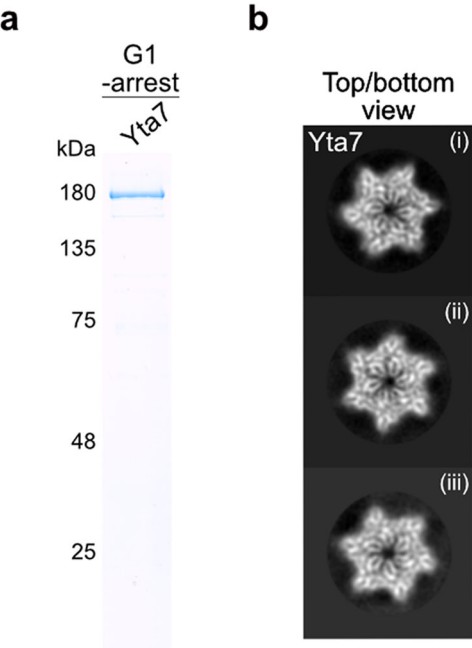

**Fig. 2 Yta7 is related to molecular segregases. a** SDS-PAGE and Coomassie staining of purified Yta7 from G1 phase cells. **b** ((i)–(iii)) Representative 2D-class averages of Yta7 purified as in **a**. Particle diameter is approximately 150 Å.

**Yta7 is related to molecular segregases**. Yta7 is distinct from classical chromatin remodelers bearing SNF2-family ATPase domains (SWI/SNF, ISWI, CHD, and INO80 families)[35–42], as it belongs to the large AAA+ family of ATPases[43,44] and contains tandem ATPase domains. This categorizes it as a type II AAA+-ATPase together with family members NSF, p97/Cdc48, and PEX1/6[20], which use the energy obtained from ATP hydrolysis to disassemble protein complexes, aggregates, or polymers of a wide range of substrates and are collectively referred to as molecular segregases[44].

To characterize Yta7 for biochemical and structural analysis, it was purified from G1 phase-arrested cells (Fig. 2a) to prevent S phase phosphorylation by CDK. Yta7 eluted as a hexamer after size exclusion chromatography (SEC) (Supplementary Fig. 3b), which is typical for AAA+-ATPases with segregase activity[44]. ATP binding was not important for hexamer formation as a Yta7 ATP-binding-mutant also eluted from SEC in a similar volume to WT (Supplementary Fig. 3a and b). The eluted peak was used for making cryo grids for electron microscopy analysis but due to a severe preferred orientation, a reliable 3D reconstruction could not be obtained. However, 2D-class averages clearly revealed a hexameric, six-fold symmetric structure typical for this family of AAA+-ATPases (Fig. 2b), consistent with previous results, revealing hexameric structures of Abo1, the Yta7 homolog in fission yeast[18]. As with Yta7, ATP binding was not necessary for Abo1′s hexamer formation[18].

**Yta7 recruits to acetylated chromatin but does not modify chromatin structure**. Having purified Yta7 in hand, we next established an in vitro method to assay for Yta7 activity. Our strategy was to set up the assay as physiological as possible. We used nucleosomal arrays on natural DNA instead of mono-nucleosomes on artificial DNA sequences that were designed to strongly bind nucleosomes[45]. We also used yeast histones purified from bacterial cells without any modification or artificial tags[5,46]. Finally, to be able to stage reactions and change buffers and

protein compositions, the DNA template was coupled to para-magnetic beads. We first reconstituted Yta7 recruitment to chromatin. Figure 3a shows the reaction scheme of the assay. Briefly, chromatin was assembled onto bead-bound DNA in the presence of the histone chaperone Nap1 and the chromatin remodeler ISW1a, ATP, and an ATP regeneration system as described previously[5]. After assembly, chromatin was extensively washed to remove Nap1 and ISW1a[5]. We could detect little recruitment of Yta7 to unmodified chromatin (Fig. 3b, lanes 1 and 2). However, H3 acetylation (H3ac) by pSAGA and acetyl-CoA, significantly stimulated Yta7 recruitment (Fig. 3b, lanes 3 and 4), hence all subsequent in vitro experiments utilized H3ac. Interestingly, Yta7 was not recruited to DNA in the absence of nucleosomes (Fig. 3c), further distinguishing it from classical chromatin remodelers, which all efficiently bind naked DNA. However, chromatin composition, as determined by immuno-blotting measuring H2A and H3 levels, was unchanged by Yta7 recruitment, suggesting that Yta7 cannot efficiently disassemble nucleosomes under these conditions (Fig. 3b).

It is possible that Yta7 associates with chromatin, maybe in a complex with partially disassembled nucleosomes. To test this, we washed the beads after Yta7 recruitment with a buffer containing high salt (0.6 M KCl). Figure 3d shows the modified reaction scheme of the assay. This high salt wash efficiently removes Yta7 but does not affect histone content (Supplementary Fig. 4a and b). Even under these stringent conditions, we could not observe a Yta7-dependent change in chromatin composition (Fig. 3e (i)). To assay for nucleosome repositioning, arrays before and after Yta7 binding were digested with low amounts of micrococcal nuclease (MNase) to generate nucleosomal ladders, which showed no change upon Yta7 recruitment Fig. 3e (ii). Taken together, these results indicate that the recruitment of Yta7 has little effect on bound chromatin. As Yta7 was purified from the G1 phase, a lack of chromatin changes may be due to the absence of phosphorylation on Yta7 residues targeted by CDK during the S phase.

**S-CDK targets Yta7 and stimulates its ATPase activity**. To explore our hypothesis that CDK could stimulate Yta7 activity, we purified S-CDK (Clb5-Cdc28-Cks1) from nocodazole treated cells (Fig. 4a) and first tested if Yta7 is a direct substrate in an in the vitro kinase assay. We also included purified Yta7 ATP-binding-mutants and Phospho-mutants from G1 phase-arrested cells (Supplementary Fig. 3a) in our assay. As with the ATP-binding mutant, the Phospho-mutant clearly formed hexameric structures (Supplementary Fig. 3b). Yta7 WT and the ATP-binding-mutant were clear substrates of S-CDK, whereas the purified Phospho-mutant protein (lacking the seven CDK consensus sites (Fig. 1b), did not show detectable phosphorylation (Fig. 4b). This phosphorylation was not caused by a contaminating kinase as it was inhibited by the S-CDK inhibitor Sic1 (Supplementary Fig. 5a). To further test if our in vitro assay is specific, we firstly observed robust phosphorylation of the Cac1 subunit of the CAF-1 complex, a chaperone involved in the deposition of histones during replication, and a known target of S-CDK[47] (Supplementary Fig. 5b and c). Second, FACT, a histone chaperone important for S phase chromatin replication and not associated with S-CDK, did not show detectable phosphorylation, although bearing CDK consensus sites in the primary structure (Supplementary Fig. 5b and c). Together, all our experiments show that the in vitro kinase assay is specific and that Yta7 is a bonafide target of S-CDK.

We next investigated if S-CDK phosphorylation of Yta7 might stimulate its ATPase function. ATPase assays of Yta7 would be confounded by the presence of S-CDK. To circumvent this issue, Yta7 was pre-phosphorylated by S-CDK. To eliminate S-CDK

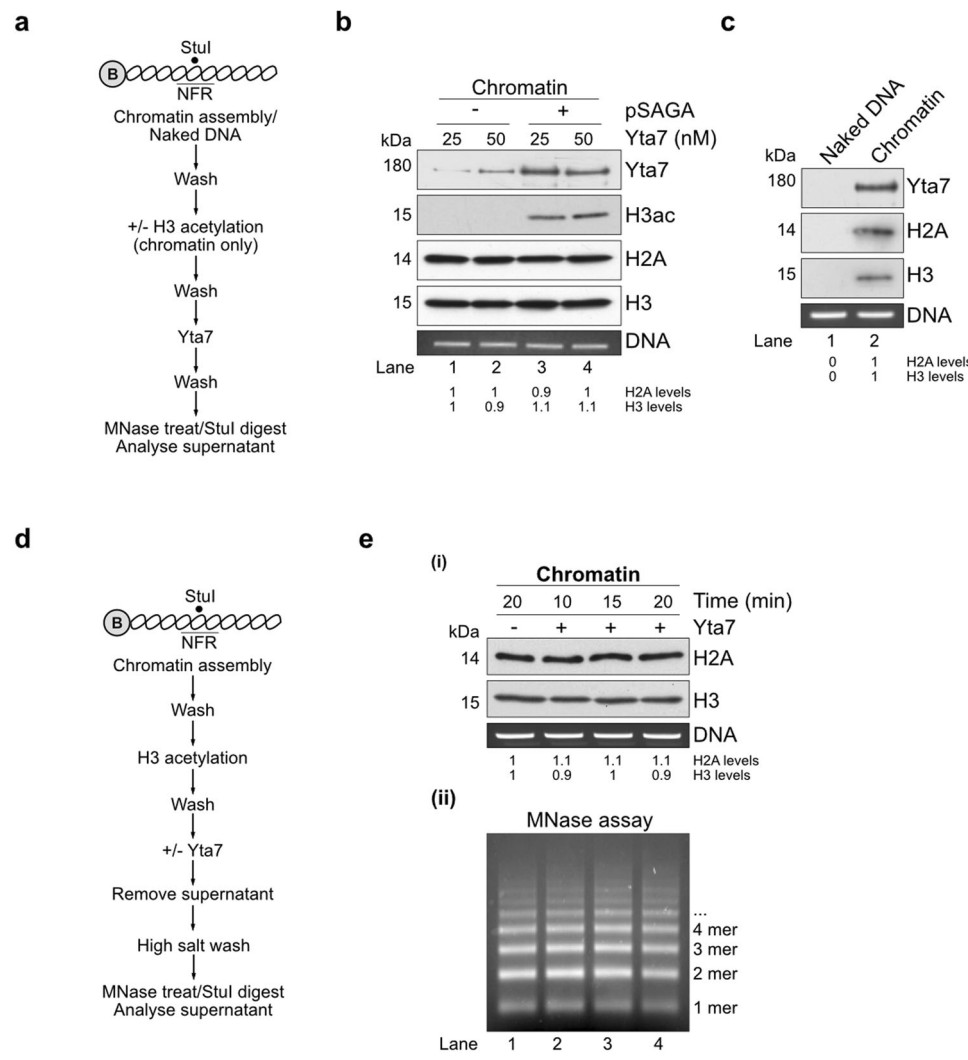

**Fig. 3 Yta7 exclusively localizes to chromatin. a** Reaction scheme of the Yta7 recruitment assay. **b** Chromatin was assembled onto 2.8 kb linear DNA coupled to paramagnetic beads. For H3 acetylation, pSAGA and acetyl-CoA were added after nucleosome assembly. H2A, H3, acetylated H3, and Yta7 were analyzed by immunoblotting. For DNA controls, chromatin templates were digested with StuI, which cuts efficiently in the nucleosome-free region (NFR) of the template[5]. Proteins were removed and DNA was analyzed by agarose gel electrophoresis and ethidium bromide (EtBr) staining. For this and all subsequent experiments involving histone immunoblots, H2A and H3 levels were quantified as described in "Methods" section. **c** Assay of Yta7 recruitment to naked DNA. The experiment was performed as in b except without chromatin assembly. **d** Reaction scheme of modified Yta7 assay determining nucleosome positioning and chromatin compositing. **e** Chromatin was assembled as in b and included H3 acetylation by pSAGA and acetyl-CoA. After Yta7 incubation, reactions were washed with high salt (0.6 M KCl). (i) Immunoblot of H2A and H3 after treatment with high concentrations of MNase or DNA digested with StuI were analyzed as in **b**. (ii) Nucleosomal positioning was determined by partial MNase digestion. After digestion, proteins were removed and the nucleosomal DNA ladder was resolved through 1.3% agarose and stained with ethidium bromide. Shown are representative experiments, which have been biologically replicated three times.

activity, samples were then treated with the S-CDK inhibitor Sic1 and phosphorylated Yta7 was re-purified to remove S-CDK. Figure 4c shows the experimental setup of these experiments. ATPase assays with pre-phosphorylated Yta7 samples were then performed in the presence of H3ac chromatin. Interestingly and consistent with our in vivo data, phosphorylation of WT Yta7 by S-CDK significantly stimulated ATP hydrolysis, but not with ATP-binding-mutants or Phospho-mutants (Fig. 4d).

**S-CDK phosphorylation regulates Yta7 chromatin segregase function to disassemble nucleosomes and to stimulate chromatin replication in vitro.** We next tested if S-CDK-dependent phosphorylation could regulate Yta7 function on chromatin. We modified our assay and phosphorylated Yta7 with S-CDK after its recruitment to chromatin followed by a high salt wash (Fig. 5a),

before determining effects on chromatin composition and nucleosome positioning. The combination of Yta7 and S-CDK resulted in a significant loss of H2A and H3, indicative of chromatin disassembly (Fig. 5b (i)). MNase treatment confirmed this and further showed that bulk nucleosomal spacing remained unaffected (Fig. 5b (iii)). Histones were not degraded as full-length H2A and H3 appeared in the supernatant after the high salt wash (Fig. 5b (ii)). These effects were not observed with Yta7-mutants ATP-binding and–Phospho-mutants or S-CDK alone (Fig. 5c (i) and (ii)) despite being efficiently recruited to chromatin and capable of forming hexameric structures, comparable to Yta7 WT (Supplementary Fig. 2b and 4c), showing the observed defects were due to defective catalysis by Yta7. This observed chromatin segregase function of Yta7 did not apply to other chromatin factors involved in chromatin replication. The histone chaperone FACT/Nhp6, as well as the chromatin

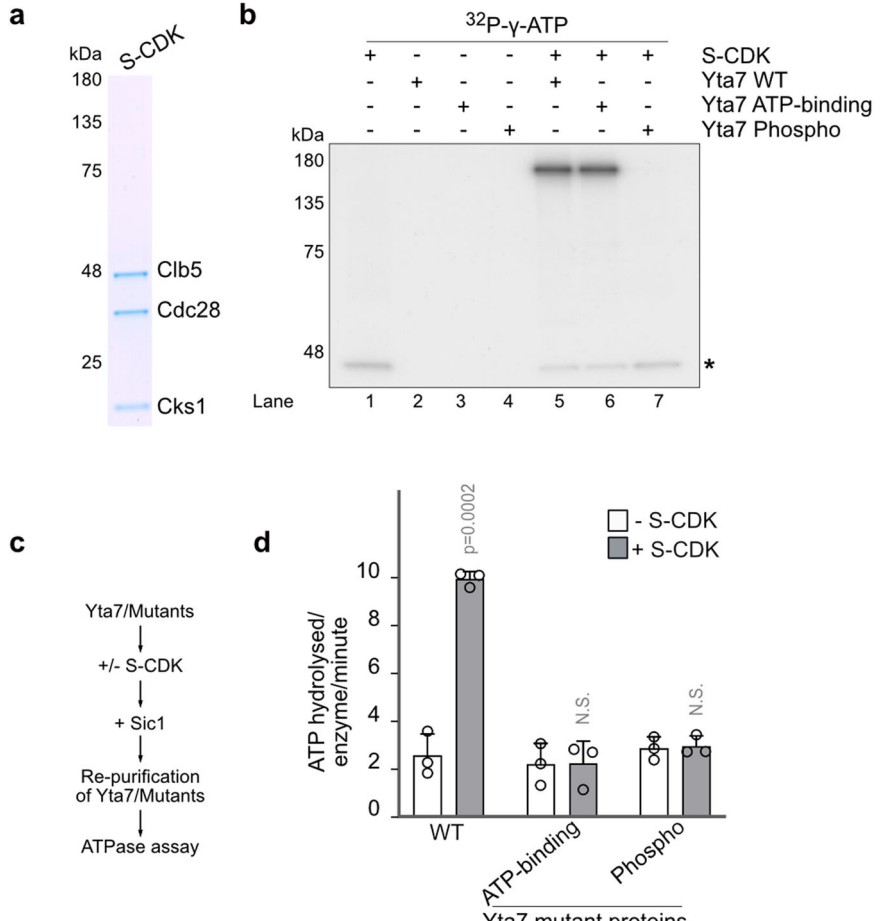

**Fig. 4 S-CDK phosphorylation stimulates ATPase activity of Yta7. a** SDS-PAGE and Coomassie staining of purified S-CDK from nocodazole-treated cells. **b** In vitro kinase assay using purified S-CDK with both WT and mutant forms of Yta7 protein (WT, ATP-binding, Phospho) (Fig. 2b, Supplementary Fig. 2b). Incorporation of [$^{32}$P-γ]-ATP into Yta7 by S-CDK was visualized using autoradiography after separation by SDS-PAGE. Asterisk shows auto-phosphorylation of S-CDK. **c** Reaction scheme of the determination of the ATPase activity of Yta7. Purified Yta7 variants were pre-phosphorylated with S-CDK (omitted for controls) and treated with inhibitor Sic1 before re-purification. **d** ATP hydrolysis rates by re-purified Yta7 WT and Yta7 ATP-binding-mutants and Phospho-mutants in the presence of H3 acetylated chromatin was obtained by measuring the oxidation of NADH, which was used to regenerate hydrolyzed ATP. Error bars depict standard errors of three independent experiments. Mean values and standard deviations (S.D.) in **d** were obtained from three biological replicates. P-values were obtained by using two-tailed unpaired t-test calculations. N.S. means statistically non-significant.

remodeler ISW1a, plus-minus S-CDK, were unable to disassemble chromatin under our assay conditions (Supplementary Fig. 6).

Considering our in vivo and in vitro results, we reasoned that S-CDK phosphorylation of Yta7 influences chromatin replication by disassembling nucleosomes. To determine if Yta7 influences DNA replication directly, we applied a fully-reconstituted in vitro replication assay containing S-CDK[5,48]. Figure 5d and Supplementary Fig. 7a show the experimental set-up of the replication reactions both on chromatin, as well as on naked DNA. Paramagnetic beads were shown to interfere with replication reactions, we, therefore, assembled and purified chromatin on soluble 10.6 kb plasmid DNA before replicating this using the soluble replication system as described previously[5,44]. Briefly, chromatin was assembled in the presence of the Origin-Recognition Complex (ORC) and then purified using S-400 spin columns. Mcm2-7/Cdt1 replicative helicase (MCM) was then loaded in the presence of ATP and the loading factor Cdc6. After phosphorylation of loaded MCM by Dbf4-Dependent Kinase Cdc7 (DDK), H3 was acetylated using pSAGA and acetyl-CoA. Naked DNA was treated with identical buffers, but without chromatin assembly and H3 acetylation. After initiation of

replication with firing and replication factors (Ctf4, Dpb11, GINS, Cdc45, Pol ε, Mcm10, Sld2, Sld3/7, Pol α, Topo I, RPA, S-CDK, Csm3/Tof1, Mrc1, Pol δ, RFC, PCNA and FACT/Nhp6), Yta7 WT, ATP-binding-mutants, and Phospho-mutants were added and reactions were incubated for the time as indicated.

Whereas Yta7 did not stimulate DNA replication on naked DNA templates (Supplementary Fig. 7b), Yta7 strongly promoted replication when templates were chromatinised (Fig. 5e). Also, Yta7 did also not stimulate replication on templates where histones were not acetylated (Supplementary Fig. 8a and b), consistent with our recruitment results (Fig. 3b). As with previous experiments, the Yta7 ATP-binding-mutants and Phospho–mutants were unable to stimulate chromatin replication (Fig. 5e). This strong effect is likely to be the result of facilitated replication origin escape of activated replisomes after Yta7 has relaxed the chromatin barrier, and is not due to indirect effects resulting from altered transcription.

## Discussion

Our results lead us to propose the following model about the mechanism and regulation of Yta7 chromatin segregase (Fig. 6).

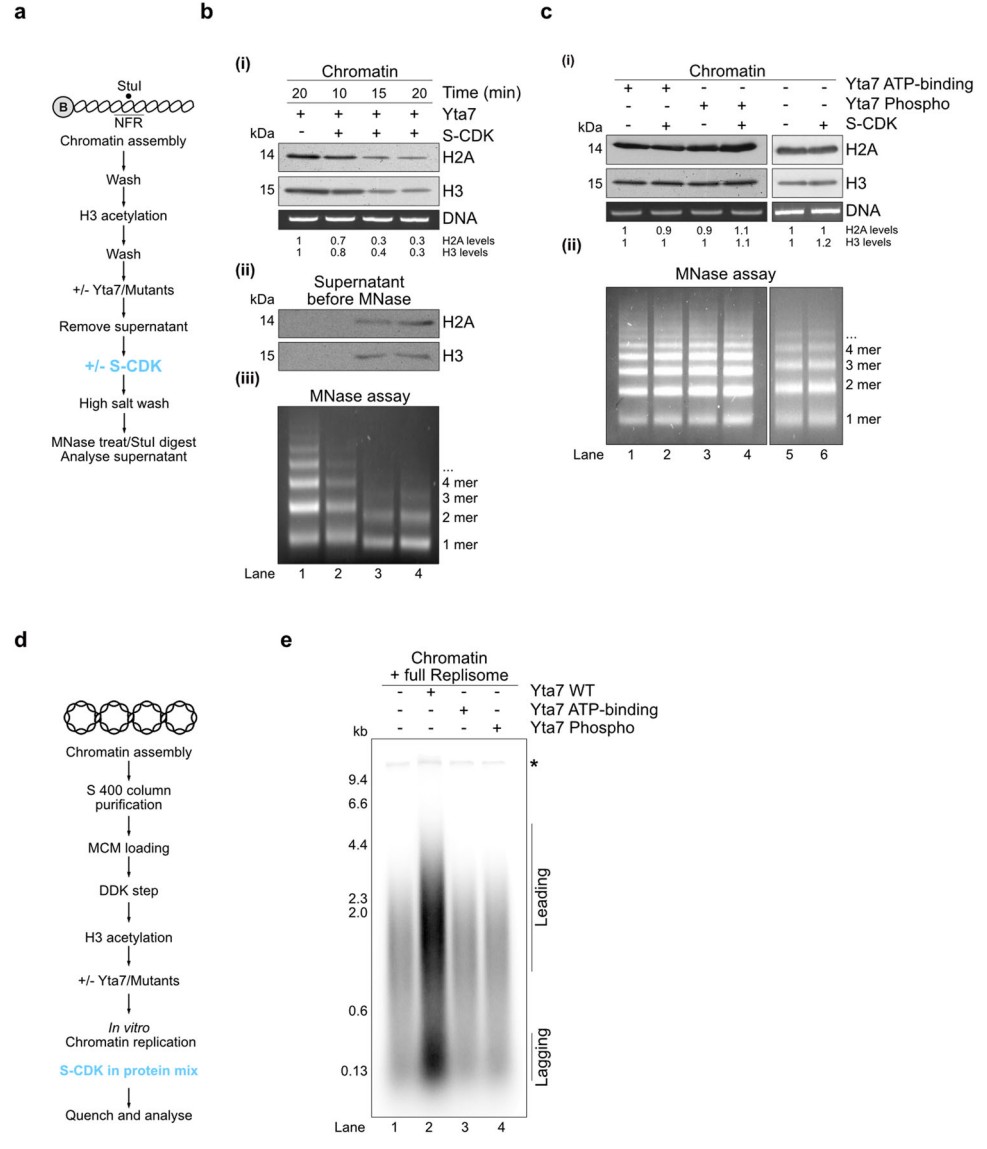

**Fig. 5 S-CDK phosphorylation of Yta7 stimulates chromatin segregase function and chromatin replication. a** Reaction scheme of the analyses of Yta7 function on chromatin. **b** Experiment as in Fig. 3a and b but with the inclusion of Yta7 phosphorylation by S-CDK. In addition, protein samples in the supernatant were analyzed by immunoblot after the first high salt wash to detect proteins released after Yta7 treatment (ii). **c** Experiments as in a and b but using mutant Yta7 proteins and an S-CDK only control. **d** Reaction scheme of the in vitro chromatin replication assay including Yta7 and Yta7 mutants. **e** Effect of Yta7 on in vitro replication. Chromatin was assembled onto ARS1-containing 10.6 kb plasmid DNA in solution as described previously[5]. After purification of chromatinised templates via gel filtration, the replicative helicase Mcm2-7/Cdt1 (MCM) was loaded, phosphorylated by Dbf4-Dependent Kinase Cdc7 (DDK) and H3 was acetylated by pSAGA and acetyl-CoA followed by the addition of Yta7 and mutant versions. Chromatin replication reactions were initiated and included Ctf4, Dpb11, GINS, Cdc45, Pol ε, Mcm10, Sld2, Sld3/7, Pol α, Topo I, RPA, S-CDK, Csm3/Tof1, Mrc1, Pol δ, RFC, PCNA, and FACT/Nhp6. Reactions were stopped after 7 min and newly replicated DNA was visualized by the incorporation of [α−32P] deoxycytidine triphosphate (dCTP) into nascent DNA. Products were separated through 0.8% alkaline agarose gels and visualized by phosphoimaging. Leading and lagging strands are visible because the assay does not include factors required for Okazaki fragment maturation. Asterisk indicates the end-labeling of nicked plasmid DNA. Shown are representative experiments, which have been biologically replicated three times.

Yta7 recruitment to chromatin is dependent on H3ac. At this stage, ATP might be bound to Yta7 but cannot be hydrolyzed efficiently. S-CDK phosphorylation of Yta7 at residues proximal to the catalytic ATPase domain stimulates ATP hydrolysis. Activation of the ATPase results in the disassembly of nucleosomes, clearing the genome of the major barrier to the replication machinery. Although a detailed mechanism is yet to be determined, an obvious candidate is that Yta7 binds and threads histone tails through the central channel of the AAA+-ATPase, and upon repeat cycles of ATP hydrolysis, translocates the histone protein through it to disassemble the parental nucleosome.

Segregated histones might then be transferred onto nascent DNA, supported by histone chaperones like FACT[49] or Mcm2[50,51], thus contribute to epigenetic inheritance[52–58]. Consistent with this idea, Yta7 was shown to physically interact with FACT, as well as with Mcm2[8,9,59]. Answering these questions is clearly an important goal in the future. A first hint that Yta7 might be involved in the positioning of histones onto nascent DNA comes from recent work, showing that Abo1, Yta7´s fission yeast homolog, is able to act as a histone chaperone[18]. Similar to Yta7, Abo1 forms hexameric structures, independent of ATP binding. Interestingly, Abo1 is capable to position histones onto DNA in

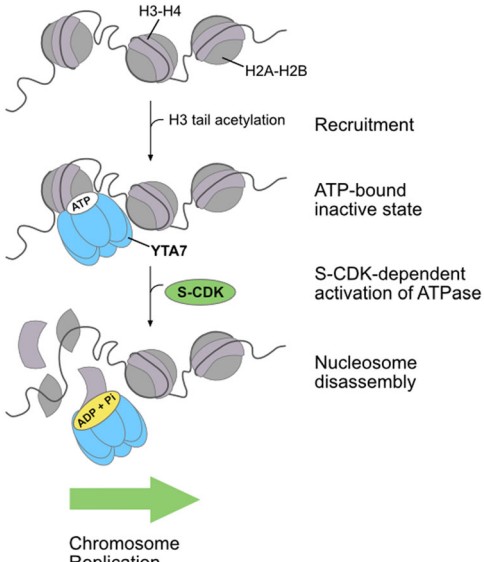

**Fig. 6 Model of Yta7 chromatin segregase mechanism and regulation.**
Recruitment of Yta7 to chromatin is facilitated by H3 acetylation. In the S phase, Yta7 is phosphorylated by CDK, which stimulates its ATPase activity. Hydrolysis of ATP is then necessary to segregate histones from DNA. This cell cycle-regulated alleviation of the chromatin barrier promotes S phase chromosome replication to preserve genome stability. See text for details.

an ATP-dependent manner and MNase protection assays displayed the presence of DNA fragments quite different from classical histone chaperones. The nature of these intermediates is not clear at the moment, but, because the authors were using non-modified histones in their assays, they might need to be acetylated. Supporting this idea, Yta7 did not promote chromatin replication on templates where histones were unmodified (Supplemental Fig. 8).

Yta7´s non-canonical bromodomain was shown to bind histones independently of posttranslational modifications and it was suggested that other regions of Yta7 might be important for chromatin association[13], thus, Yta7 recruitment may be due to acetylation-induced unfolding of nucleosome fibers rather than direct binding to the acetylated H3 tail. The dependence of acetylation for recruitment of Yta7 is corroborated by in vivo experiments showing that *YTA7* was important for DNA synthesis in the S phase under "hyperacetylated" conditions[60] as all *yta7* mutants had compromised S phase progression when combined with a deletion of histone deacetylase *RPD3* (Supplementary Fig. 9a), without activating the replication checkpoint or reducing S-CDK function (Supplementary Fig. 9b). It was shown that that dynamic histone acetylation around origins of replication promotes origin function[61], however, the molecular mechanism of this stimulation remains unknown. Our data now suggest that Yta7 is important for this.

The chromatin segregase activity of Yta7 is likely to influence other genomic processes, namely transcription[12,15]. We propose that other kinases might regulate transcription via phosphorylation of Yta7 outside of the S phase, such as Casein Kinase 2, which was shown to physically interact with Yta7[9]. The coordination of the cell cycle with chromatin segregase activity during the S phase raises the possibility that other chromatin factors involved in chromosome replication might behave similarly.

Certain oncogenes can influence S phase initiation and thereby cause replication stress[62]. How they accomplish this is still poorly understood. The human homolog of Yta7, ATAD2 is an emerging

oncogene and overexpressed in various cancers with poor prognosis[11,17,26–28]. Yet, the molecular mechanisms of ATAD2 actions are not known. As with Yta7, a cluster of phospho-sites, some of which are CDK-consensus sites, was found in close proximity of the ATPase domain of ATAD2[63–65]. In addition, ATAD2 was shown to be recruited to active replication sites, suggestive of involvement in chromosome replication[66]. Interestingly, ATAD2 was found to associate with histone modifications found in newly synthesized histones, suggesting that Yta7 might also be involved in the assembly of new histones into chromatin behind the replication fork.

Because of the high level of conservation, we suggest that ATAD2 might function in a similar manner to Yta7 and propose a model of how oncogenes might influence replication by directly manipulating chromatin structure for efficient origin opening, timing, and coordination of genome replication in the S phase to preserve genome stability.

## Methods

**Strains and oligos**. A list of strains and primers for ChIP assays used in this study can be found in Supplementary Information.

**Genetic interaction profile similarity sub-network for *YTA7***. Genes that exhibited similar genetic interaction profiles (PCC > 0.13) to *YTA7* were extracted from the global genetic interaction network[30] and laid out using a spring-embedded layout algorithm.

**Cell cycle arrest and release experiments**. Cells were grown at 30 °C in YPD to $OD_{600nm}$ of 0.5–0.6 and then arrested in G1 with alpha-factor (synthesized at the MPI core facility, 5 mg/mL stock added 1:1000) for 120 min in total. A second dose of alpha-factor was added after 60 min. Cells were then washed with pre-warmed YPD once and released into fresh pre-warmed YPD to resume the cell cycle. Samples for FACS analyses, as well as for immunoblotting were taken at indicated time points and processed according to standard protocols. Briefly, FACS samples were digested with RNase A (R 4875, Sigma) and proteinase K (P 2308, Sigma). After staining with SYTOX green (S 7020, Invitrogen; used 1:10.000), DNA content was measured on a MACSQuant analyzer 10 (Miltenyi Biotec).

**Analyses of Rad52 foci and genetic analyses of recombination**. Rad52 foci were counted in >200 S/G2 phase cells, which were transformed with pWJ1344[33]. Cells were grown in synthetic complete (SC) liquid media with 2% glucose as the carbon source. Cells were visualized using a Leica DC 350F fluorescent microscope. The mean and SEM of three experiments performed with independent transformants were plotted.

Recombination frequencies were calculated by transforming the indicated yeast strains with the pTINV-FRT plasmid containing a *leu2* inverted-repeats recombination system. Recombination tests were performed in mid-log phase cultures after growth at 30 °C in liquid SC media with 2% raffinose and 5 mg/mL doxycycline but lacking uracil. Leu⁺ recombinants were selected on SC media with 2% glucose lacking leucine and uracil. For each transformant, the median value of six independent colonies was obtained. The mean and SEM of at least three independent experiments performed with independent transformants were plotted.

**Pull-down assay of FLAG-tagged Yta7**. Cells (OD at 600 nm ~1) were harvested, frozen in liquid nitrogen, and resuspended in 1 mL lysis buffer (50 mM Tris-HCl, pH 7.2, 150 mM NaCl, 1.5 mM MgOAc, 0.15%Nonidet P-40, 5 mM EDTA, and protease inhibitors (100 mM PMSF and Complete protease inhibitor tablets, Roche)). Cells were broken by bead-beating with glass beads, extracts were collected and soluble fractions were isolated by centrifugation (30 min at 21.130 × g, 4 °C). Extracts were treated with 800 units of Benzonase (Merck, 71206-3) for 10 min at 4 °C. FLAG-tagged Yta7 was pulled down with anti-FLAG M2 affinity gel (Sigma) in batch (2 h at 4 °C with rotation). After three washing steps in lysis buffer (5 min at 4 °C), Yta7-FLAG and physically associated proteins were eluted with 0.25 mg/mL 3× FLAG peptide with shaking for 1 h at 4 °C. Supernatants were collected and interacting proteins were analyzed by immunoblotting.

**ChIP assays**. Cells bearing TAP-tagged Yta7 were arrested with alpha-factor and released as described in "Cell cycle arrest and release experiments". Cells were harvested at 15 min after release from alpha-factor block (early S phase, Fig. 1c). Immunoprecipitated DNA was analyzed by qPCR. For all experiments, internal control was included (a non-transcribed region of chromosome V). ChIP efficiency means the ratio of specific primer/control for immuno-precipitation divided by the same ratio for input.

**Protein expression and purification**. Yta7 expression and purification—Codon optimized versions of wild-type Yta7, as well as an ATPase mutant and mutant lacking S-CDK phosphosites were chromosomally integrated and overexpressed in *S. cerevisiae* from a GAL1-10 promoter in a *pep4Δ, bar1Δ* background (see Supplementary Information for details on the expression strains). All three constructs had a 3xFLAG tag on the C-terminus, which was used for the first step of protein purification, as well as for the detection in subsequent assays by immunoblotting (see below). For purification of mutant versions of Yta7, the endogenous copy was eliminated.

To induce expression, cells were grown in YP media containing 2% raffinose as the carbon source at 30 °C with shaking to a density of ~2–4 × 10⁷ per ml. For subsequent assays, it was imperative to purify Yta7 and mutant versions that are not phosphorylated by S-CDK. To achieve this, strains were arrested in G1 with 100 ng ml⁻¹ alpha-factor (GenScript) and incubation was continued for 3 h. Protein expression was induced by the addition of 2% galactose for 3 h at 30 °C. Cells were then collected and washed twice with 25 mM HEPES-KOH pH 7.6, 1 M sorbitol and once with lysis buffer (250 mM KCl, 25 mM Tris-HCl pH 7.2 (25 °C), 10% glycerol, 0.05% NP-40, 1 mM EDTA, 4 mM MgCl₂) without protease inhibitors. Cells were then resuspended in an equal volume of 2× lysis buffer plus protease inhibitors and the suspensions were frozen drop-wise in liquid nitrogen. The frozen cells were crushed using a freezer mill (SPEX CertiPrep 6850 Freezer/Mill) (6 cycles for 2 min, crushing rate 15). The resulting powders were collected and stored at −80 °C until further use.

For protein purifications, the powder was slowly thawed on ice before adding an equal amount of lysis buffer plus protease inhibitors (cOmplete, Roche), 7.5 mM benzamidine, 0.5 mM AEBSF, 1 mM pepstatin A, 1 mg/mL protinin (Sigma), and 1 mM leupeptin (Merck)). Insoluble material was cleared by centrifugation (235.000 × g, 4 °C, 1 h) and Yta7-FLAG₃ₓ and mutant versions were bound to pre-washed anti-FLAG M2 affinity gel (Sigma) in batch for 45 min at 4 °C. After transferring into a disposable column (Bio-Rad) beads were extensively washed with lysis buffer without protease inhibitors to avoid co-purification of contaminating kinase (200–300 CV). Cells were then washed with 100 CV of lysis buffer containing 150 mM KCl. Yta7 and mutant versions were eluted in 1 CV of lysis buffer containing 150 mM KCl with 0.5 mg/mL 3× FLAG peptide (Sigma), followed by 2 CV of lysis buffer containing 150 mM KCl with 0.25 mg/mL 3× FLAG peptide. The eluates were pooled and further purified using a 1 mL MonoQ column. Yta7 and mutant versions were eluted with a 20 CV gradient from 0.1 to 1 M KCl in lysis buffer including 1 mM DTT. Peak fractions were analyzed by SDS-PAGE and desired fractions were dialyzed against 2 liters of 150 mM KCl, 25 mM Tris-HCl pH 7.2 (25 °C), 40% glycerol, 0.05% NP-40, 1 mM EDTA, and 4 mM MgCl₂. Size Exclusion Chromatography was performed prior to cryo-EM analysis using a Superose 6 (3.2/300) column in 150 mM KCl, 25 mM Tris-HCl pH 7.2 (25 °C), 5% glycerol, 0.05% NP-40, 1 mM EDTA and 4 mM MgCl₂. Protein concentrations were determined using a Bradford reagent (Bio-Rad).

Sic1 expression and purification—1 L of *E. coli* (BL21 Codon plus DE3 RIL) plus Sic1-His₆ expression vector were grown at 37 °C to a density of OD₆₀₀ₙₘ of 0.6. Cells were then chilled on ice for 20 min before 1 mM IPTG was added. Protein expression was induced overnight at 16 °C. After harvesting, cells were resuspended in 30 mL 2× lysis buffer (50 mM HEPES-KOH pH 7.6, 0.04% NP-40, 20% glycerol, 300 mM NaCl, 2 mM ß-Mercaptoethanol, protease inhibitors (cOmplete, Roche), lysozyme was added (60 mL (50 mg/ml)) and rotated for 20 min at 4 °C. Cells were then broken by sonication (2 min, 40% output) and insoluble material was collected by centrifugation (27.000 × g at 4 °C, JA-25.50). The cleared lysate was incubated with washed Ni-NTA beads plus 10 mM imidazole for 1 h at 4 °C. Beads were washed twice with 1× lysis buffer plus 10 mM imidazole before loading onto a disposable column (Bio-Rad). After two washes (lysis buffer plus 25 mM imidazole), proteins were eluted using lysis buffer plus 200 mM imidazole. Fractions were analyzed using SDS-PAGE and peak fractions were collected and further purified using a 1 mL MonoQ column (GE Healthcare). Sic1 was eluted with 20 column volumes (CV) of a 0.1 to 1 M KCl gradient (25 mM HEPES-KOH pH 7.6, 0.02% NP-40, 10% glycerol, 0.1/1 M KCl, 1 mM DTT). Peak fractions were analyzed by SDS PAGE, pooled, and concentrated. Sic1 was further purified using a Superdex200 10/300 GL gel filtration column (25 mM HEPES-KOH pH 7.6, 0.02% NP-40, 10% glycerol, 0.2 M K-glutamate, 1 mM DTT). Peak fractions were pooled, concentrated, flash-frozen, and stored at −80 °C.

CAF-1 expression and purification—Cell powder was resuspended in 100 mM NaCl, 10 mM MgCl₂, 0.01% NP-40, 25 mM Tris-HCl pH 8.0, 10% glycerol, 1 mM DTT + protease inhibitors. Insoluble material was cleared by centrifugation at 235.000 × g for 45 min at 4 °C. CaCl₂ was added to the extract at a final concentration of 3 mM along with 1 mL of calmodulin affinity resin and rotated for 90 min at 4 °C. The resin was collected and washed with 50 CV of buffer containing 100 mM NaCl, 10 mM MgCl₂, 0.01% NP-40, 25 mM Tris-HCl pH 8.0, 10% glycerol, 1 mM DTT, 2 mM CaCl₂, and the protein was eluted with 10 CV of buffer containing the same buffer plus 4 mM EGTA and 2 mM EDTA. The eluate was pooled and further purified using MonoQ. CAF-1 was eluted over a 30 CV gradient from 100 mM NaCl to 1 M NaCl. Peak fractions were pooled, concentrated, and applied to a Superdex 200 Increase 10/300 GL column (GE Healthcare). Fractions containing CAF-1 were then pooled, concentrated, and stored in aliquots at −80 °C until further use.

**Chromatin assembly, H3 acetylation, and Yta7 recruitment on bead-coupled linear DNA**. A biotin-labeled linear 2.8 kb DNA fragment containing the ARS1 origin sequence was prepared, coupled to magnetic streptavidin-coated beads (Dynabeads M-280, Invitrogen), and chromatin assembly was carried on. Yeast histones (370 nM), Nap1 (3.5 mM), and ISW1a (5 nM) as well as an ATP regenerating system (creatine phosphate (40 mM), ATP (3 mM), and creatine phosphate kinase (0.6 mL in 40 mL of a 14 mg/mL stock solution)) were incubated with 500 ng of bead-coupled DNA for 4 h at 30 °C with shaking (1250 rpm). The chromatin assembly buffer was 90 mM KOAc, 25 mM HEPES-KOH pH 7.6, 10 mM Mg(OAc)₂, 0.02% NP-40, 1 mM DTT, 5% glycerol and 0.1 mg/mL BSA. For recruitment assays using H3 acetylated chromatin as a template, unincorporated histones were removed by washing beads twice with wash buffer containing 300 mM K-glutamate, 25 mM HEPES-KOH pH 7.6, 10 mM Mg(OAc)₂, 0.02% NP-40 and 1 mM DTT and once with wash buffer plus 100 mM K-glutamate. For H3 tail acetylation, 300 nM pSAGA and 20 mM acetyl-CoA were added for 30 min at 30 °C with shaking (1250 rpm) (100 mM K-glutamate, 25 mM HEPES-KOH pH 7.6, 10 mM Mg(OAc)₂, 0.02% NP-40, 1 mM DTT and 0.1 mg/mL BSA)[5]. For assays using naked DNA templates, DNA coupled to beads was treated with the same buffers but without histones, ISW1a, Nap1, pSAGA, ATP, and acetyl CoA.

For Yta7 recruitment assays, chromatinised templates were first washed with binding buffer containing high salt (600 mM NaCl), which removes excess histones, ISW1a, Nap1, and pSAGA but does not affect chromatin (Fig. 3a, Extended data Fig. 6a). Templates were then washed once in binding buffer without ATP and BSA. Beads were resuspended in binding buffer plus BSA (0.1 mg/mL) and 5 mM ATP. Yta7 was added at indicated concentrations and incubated at 30 °C for 30 min with shaking (1250 rpm). Beads were then washed with a binding buffer (0.3 M K-glutamate (or 100, 200, 400, and 600 mM NaCl (Extended Data Fig. 6a)) without ATP and BSA before being resuspended in binding buffer containing 100 mM K-glutamate. The reactions were then split in half—one half was used for analyzing DNA content as a loading control, as well as to exclude possible nuclease contaminations of Yta7 preparations, which was imperative for the accurate interpretation of all subsequent assays. The nucleosome-free region (NFR) around the ARS1 origin bears a StuI site for restriction digest (ref. [5]; Fig. 2d). After adding StuI (New England Biolabs, NEB), samples were incubated in the corresponding buffer (CutSmart, NEB) for 1 h at 37 °C with shaking (1250 rpm). Beads were then collected on a magnetic rack, released DNA was purified from protein components (High pure PCR Product Purification Kit, Roche) and analyzed using ethidium bromide (EtBr) agarose gel electrophoresis.

To determine Yta7 recruitment, histone content, as well as acetylated H3, chromatin, and chromatin-bound fractions, were released from magnetic beads using a binding buffer containing 5 mM CaCl₂ plus an excess (2000 units/20 mL reaction) of micrococcal nuclease (MNase; NEB, M0247S) Beads were incubated for 5 min at 37 °C with shaking (1250 rpm), collected as above and the supernatant was analyzed by immunoblotting. Immunoblots were quantified using ImageJ. Briefly, a rectangular area of the same size was measured for all lanes. The resulting values were normalized first to the lane with the control experiment (which was set to 1). In a second step, results were normalized to the DNA control.

**Cryo-electron microscopy of Yta7**. Purified Yta7 (0.7 mg/mL) was applied to UltrAuFoil Holey Gold grids that were glow discharged and vitrified in liquid ethane using a Vitrobot Mark IV with a 5.5 s blotting time. Data were acquired using a Titan Krios microscope (Thermo Fisher) operated at 300 KeV equipped with a K3 detector (Gatan) and energy filter with 20 eV slit, located at the Electron Bio-Imaging Centre at Diamond Light Source. The detector was operating in counting mode with a pixel size of 0.845 Å. The total dose was 50 electron/Å² over 50 frames. Data were collected using EPU software with a nominal defocus range set from −1.5 to −3.5 microns. Movie frames were motion-corrected using MotionCor2 and contrast transfer functions estimated with CTFFIND4. Particle picking was attempted with Cryosparc, Gautomatch, Warp, and crYOLO, with particle numbers ranging from 30k to 85k. A severe preferred orientation was observed, as 2D classification in either RELION or Cryosparc resulted in classes that represented a single orientation of Yta7, corresponding to a view along its six-fold symmetry axis.

**In vitro S-CDK reactions**. Twenty-five nanomolar of Yta7 and mutant versions, FACT and CAF-1 were incubated with 5 nM S-CDK in buffer containing 100 mM K-glutamate, 25 mM HEPES-KOH pH 7.6, 10 mM Mg(OAc)₂, 0.02% NP-40, 1 mM DTT, 10 mM ATP and 5 mCi ³²P-g-ATP for 30 min at 30 °C. For reactions including Sic1, S-CDK was pre-incubated with indicated concentrations of Sic1 for 30 min. Excess, unbound Sic1 was removed by re-binding of S-CDK/Sic1 complexes to calmodulin affinity resin and purified. Proteins were then separated on SDS-PAGE, gels were dried and exposed with Super RX Medical X-Ray Film (FUJI) or scanned using a Typhoon phosphoimager (GE Healthcare).

**Pre-phosphorylation of Yta7**. One hundred nanomolar of FLAG-tagged Yta7, as well as mutant versions were incubated with 25 nM S-CDK (Clb5-Cdc28) in buffer containing 100 mM K-glutamate, 25 mM HEPES-KOH pH 7.6, 10 mM Mg(OAc)₂, 0.02% NP-40, 1 mM DTT, 0.1 mg/mL BSA and 5 mM ATP for 30 min at 30 °C.

Sic1 (250 nM) was then added and reactions were incubated for another 3 min at 30 °C. Samples were then diluted in the same buffer (100 mM K-glutamate) without ATP and incubated with magnetic anti-FLAG M2 beads (Sigma, M8823) for 45 min at 4 °C with rotation. Beads were then washed 6 times with buffer as above plus 300 mM K-glutamate and without ATP and once with buffer containing 150 mM KCl, 25 mM Tris-HCl pH 7.2 (25 °C), 10% glycerol, 0.05% NP-40, 1 mM EDTA, and 4 mM MgCl$_2$. Proteins were eluted in the same buffer supplemented with 0.25 mg/mL 3× FLAG peptide with shaking for 1 h at 4 °C. The supernatant was collected, aliquoted, and flash-frozen in liquid nitrogen for −80 °C storage. Protein concentrations were determined using a Bradford reagent (Biorad) and equal protein amounts were additionally verified by silver staining (Invitrogen).

**NADH oxidation-coupled ATPase assay**. Twenty-five nanomolar of the proteins were incubated in buffer containing 50 mM NaCl, 25 mM HEPES-KOH pH 7.6, 0.1 mM EDTA and 10% glycerol, supplemented with 2.5 mM ATP (Sigma, A 3377), 2.5 mM MgCl$_2$, 0.4 mM NADH (Sigma, N4505), 3 mM phosphoenolpyruvate (PEP) (Molecula, 16921512) and 16 units/mL lactic dehydrogenase/pyruvate kinase enzymes (Sigma, P0294). 30 mL reactions were prepared in 384 well plates (Greiner, 781101) in buffer without ATP and MgCl$_2$. Each reaction contained 6 pmol of chromatin acetylated on H3 tails. To prepare this, chromatin was assembled onto soluble 10 kb plasmid DNA templates. H3 acetylation was carried out as in "Chromatin assembly, H3 acetylation and Yta7 recruitment on bead-coupled linear DNA". Excess histones and enzymes were removed using an S-400 spin column (GE Healthcare). ATPase reactions were initiated by adding an ATP/MgCl$_2$ Mix (final concentration 2.5 mM each). Plates were incubated at 26 °C in a Biotek PowerWave HT plate reader for 60 min and absorption at 340 nm was determined every 15 s. For analyses, each time course was fitted to a linear function within a time range where all reactions were linear. From the slope of the reaction and the extinction coefficient of NADH (6220 M$^{-1}$ cm$^{-1}$), the change in NADH concentration was calculated (for 30 mL reactions in Greiner plates, the path length was 0.272727 cm). As oxidation of 1 NADH equals the hydrolysis of 1 ATP, the ATP hydrolysis rates were calculated from the slopes.

**Chromatin-segregase assay**. To determine chromatin disassembly activity of Yta7, chromatin was first assembled onto DNA coupled to magnetic beads as described in "Chromatin assembly, H3 acetylation and Yta7 recruitment on bead-coupled linear DNA". As for recruitment assays, templates were first washed with binding buffer containing high salt (600 mM NaCl), to remove ISW1a, Nap1, and pSAGA. Chromatin templates were then washed once in binding buffer without ATP and BSA and then resuspended in binding buffer plus BSA (0.1 mg/ml) and 5 mM ATP. 25 nM Yta7 and mutant versions were added and incubated at 30 °C for 10 min with shaking (1250 rpm). After collecting beads on a magnetic rack, the supernatant was carefully removed and chromatin templates with bound Yta7 and mutant versions were resuspended in a binding buffer containing 5 mM ATP and BSA (0.1 mg/mL). Twenty-five nanomolar S-CDK was added and samples were incubated for the time indicated with shaking (1250 rpm/min) at 30 °C. Beads were then collected and washed with buffer containing 600 mM NaCl, 25 mM HEPES-KOH pH 7.6, 10 mM Mg(OAc)$_2$, 0.02% NP-40, and 1 mM DTT. For examining histone content in the supernatant (Fig. 4a (ii), "Supernatant before MNase"), the first high salt wash was collected, proteins precipitated using trichloroacetic acid and analyzed by immunoblotting as described above. Otherwise, the reactions were washed again in a high salt buffer and then in a buffer containing 100 mM K-glutamate. Reactions were then treated with StuI to control for DNA content and possible nuclease contaminations or with high amounts of MNase (2000 units/20 mL reaction) to release chromatin from magnetic beads as described in "Chromatin assembly, H3 acetylation and Yta7 recruitment on bead-coupled linear DNA". In addition, to determine nucleosome spacing on bulk chromatin, reactions (60 mL) were treated with 100 units MNase for 6 min at 37 °C in buffer containing 100 mM K-glutamate and 5 mM CaCl$_2$ (1250 rpm). Beads were then collected, released DNA was purified from protein components (High pure PCR Product Purification Kit, Roche) and chromatin quality-analyzed using EtBr agarose gel electrophoresis (1.3% Biozym ME Agarose, 840014). Histone levels were normalized as in "Chromatin assembly, H3 acetylation and Yta7 recruitment on bead-coupled linear DNA".

**In vitro chromatin replication reactions**. For replication reactions on chromatinised templates, chromatin was assembled on 10.6 kb of plasmid DNA (1.3 nM) bearing the yeast replications origin ARS1 in the presence of 60 nM ORC. To remove excess soluble histones and other components, reactions were put over a MicroSpin S-400 HR column (GE Healthcare), which was equilibrated three times with 250 mL buffer containing 25 mM HEPES-KOH pH 7.6, 100 mM K-glutamate, 10 mM Mg(OAc)$_2$, 0.02% NP-40 and 1 mM DTT. All subsequent reactions were carried out at 30 °C. After a MCM loading step for 30 min (50 nM Mcm2-7/Cdt1; 80 nM Cdc6 in buffer containing 100 mM K-glutamate, 25 mM HEPES-KOH pH 7.6, 10 mM Mg(OAc)$_2$, 0.02% NP-40, 1 mM DTT in the presence of 5 mM ATP), 50 nM DDK was added and the reactions incubated for another 30 min. H3 acetylation was then carried out as described above (20 mM acetyl CoA, 300 nM pSAGA, 30 min). Protein amounts were 20 nM Ctf4, 30 nM Dpb11, 220 nM GINS, 40 nM Cdc45, 20 nM Pol ε, 5 nM Mcm10, 50 nM Sld2, 25 nM Sld3/7, 60 nM Pol α,

10 nM TopoI, 100 nM RPA, 20 nM S-CDK, 20 nM Csm3/Tof1, 10 nM Mrc1, 10 nM Pol δ, 20 nM RFC, 20 nM PCNA, 25 nM FACT and 250 nM Nhp6. Yta7 and mutant versions were added at 25 nM. The buffer conditions were 200 mM K-glutamate, 25 mM HEPES KOH pH 7.6, 10 mM Mg(OAc)$_2$, 100 mg/mL BSA, 1 mM DTT, 0.01% NP-40, 3 mM ATP, 60 nM [α$^{32}$P]-dCTP, 20 mM dGTP, dATP, dTTP, and 5 mM dCTP 160 mM CTP, UTP, GTP.

Replication reactions on naked DNA were performed as in ref. [5]. As above, Yta7 and mutant versions were added at 25 nM. All reactions were stopped using 30 mM EDTA and unincorporated nucleotides were removed using Illustra MicroSpin G-50 columns (GE healthcare). Samples were run through 0.8% alkaline agarose gels including 30 mM NaOH and 2 mM EDTA for 16 h at 24 V. After fixing with 20% trichloroacetic acid, gels were dried onto Whatman papers and scanned using a Typhoon phosphoimager (GE Healthcare).

**Antibodies**. Antibodies for immunoblots were anti-Rad53 (Abcam, ab 104232) dilution: 1:2000, anti-Orc6 (SB49, a kind gift from Steve Bell) dilution: 1: 3000, anti-Mcm7 (Santa Cruz, yN-19, Sc 6688) dilution: 1:2000, anti-FLAG M2 peroxidase (Sigma, F 3165) (for Yta7/Mutants-FLAG$_{3x}$) dilution 1: 5000, anti-TAP (Thermo Fisher, CAB1001) dilution 1: 5000, anti-acetyl histone H3 (Millipore 06-599) dilution 1: 2000, anti-H2A (Active Motif, 39235) dilution 1: 2000 and anti-H3 (Abcam, ab 1791) dilution 1: 5000.

**Reporting summary**. Further information on research design is available in the Nature Research Reporting Summary linked to this article.

## Data availability
The data that support this study are available from the corresponding author upon reasonable request. Source data are provided with this paper.

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

## Acknowledgements

We wish to thank Michael Costanzo with help in generating global genetic interaction similarity networks, John Diffley for the Sic1 expression plasmid, Steve Bell for anti-Orc6 antibody, and Silvia Härtel for technical assistance. We thank Peter Becker and Julia Kurat for critically reading the manuscript. This work was funded by the Deutsche Forschungsgemeinschaft (DFG, German Research Foundation)–Project-ID 213249687–SFB 1064 to C.F.K., B.P., and F.-M.P.; MU 3613/3-1 to F.-M.P.; PF794/5-1 to B.P. A.C.M.C. was supported by the Wellcome Trust (102535/Z/13/Z). We acknowledge Diamond for access and support of the Cryo-EM facilities at the UK national electron bio-imaging center (eBIC), proposal EM20287-21, funded by the Wellcome Trust, MRC and BBSRC. Research in A. A. lab was supported by the Spanish Ministry of Economy and Competitiveness (BFU2016-75058-P). B.G.-G. was funded by the Spanish Association Against Cancer (AECC).

## Author contributions

C.F.K. conceived the concept and experiments of the paper. E.C. and P.B. performed most of the biochemical experiments and prepared protein samples for electron microscopy, which was performed by L.M.D-S. and A.C.M.C. and A.C.M.C. aligned AAA+-ATPase sequences. In vivo cell-cycle experiments were carried out by K-U.R. and DNA repair analyses were performed by P.O. P.V. performed ATPase reactions. B.G-G., F. M-P., A.A., A.C.M.C., B.P. and C.F.K. were involved in interpreting and discussing data and provided funding. C.F.K. wrote the paper and all authors edited it.

## Funding

## Competing interests

The authors declare no competing interests.
