## [Peer Review File · Nature Communications]

A CDK-regulated chromatin segregase promoting chromosome replicationREVIEWER COMMENTS

Reviewer #1 (Remarks to the Author):

This paper shows that budding yeast lacking the AAA+ATPase protein Yta7 have a minor (5 minute) delay in progression through S-phase, and that they have more DNA damage and more recombination (Fig. 1). Yta7 sequence is similar to that of segregases and is a hexamer (Fig. 2). In vitro, Yta7 binds to chromatin rather than naked DNA, and this is enhanced by histone acetylation (Fig. 3). In vitro, Yta7 can be phosphorylated by CDK from S phase cells and this increases its ATPase activity in vitro (Fig. 4). Yta7 from S phase cells can displace histones from reconstituted chromatin and can promote replication of chromatinized templates in vitro (Fig. 5). The conclusion from these data is made that the Yta7 is a chromatin segregase promoting chromatin replication. This is an interesting idea, but needs more experimental data to support that this is the physiological role of Yta7 rather than an in vitro artifact coupled with the in vivo consequences of the known increased histone expression that occurs in yeast lacking Yta7.

Major concerns:

1. There is a large body of literature that shows that Yta7 is required to repress histone expression during phases of the cell cycle other than S. As a consequence, without Yta7, there is excess histones in the cells and excess histones on a chromatin. These facts are not even mentioned in this paper! The excess histones in the Yta7 could be the indirect explanation for the slower S-phase and more DNA damage, given that the Gunjan / Verreault labs have a large body of publications showing that overexpression of histones have these effects.
2. There is no demonstration that Yta7 localizes to chromatin in S phase in vivo or to sites of DNA replication, which would be critical to rule out such an indirect role for Yta7's influence on replication and genome stability.
3. Yta7 has previously been proposed to be a histone disassembly factor and this should be discussed (discussed in the review by Cattaneo et al., *Molecules and Cells* 2014).
4. Others have published that in mid S phase, Yta7 is heavily phosphorylated by Cdk1 and CK2, which is required for Yta7 dissociation from histone gene promoters. This should be discussed. Is this the same phosphorylation that they are seeing here by CDK – what do they even mean by CDK here actually?
5. The kinase reactions need to include more controls. Firstly another protein with serines that is not normally phosphorylated by S-CDK, to show they are not just adding excess kinase that would force the phosphorylation of anything. To conclude it is specific to S-phase CDK as they do, they need to show that CDK from other phases of the cell cycle does not have this effect.
6. To support their conclusions, they need to show that Yta7 does not get phosphorylated in S phase upon inactivation of CDK in vivo.
7. For the histone removal experiments in Fig. 5 that lead to replication, they need to show that this is unique to Yta7 and not to other chromatin remodelers or histone chaperones, because such experiments are prone to biochemical artifacts. Add enough of any histone binding protein / molecule and it will pull histones off of the DNA. What is the stoichiometry of Yta7 to nucleosomes in these experiments?

Other concerns:

1. The chromatin templates in Fig. 3B have hugely varying ratios of DNA to H3 to H2A. These experiments need to be made more consistent.

Reviewer #2 (Remarks to the Author):

This study demonstrates uses predominantly in vitro approaches to show that the *S. cerevisiae* Yta7 AAA+ ATPase is able evict histones from chromatin in vitro and that the activity of this enzyme is controlled by phosphorylation. These are significant findings that provides important insight to function of this class of chromatin remodeler and given the role of its human homolog (ANCCA/ATAD2) in cancer will be of general interest.

There are however, important issues that need to be addressed and the manuscript needs extensive modification. A major problem is that some of conclusions/claims are exaggerated and not supported by the data. Another problem is the degree to which data from previous studies from other groups are marginalized or ignored. At best this is an inadvertent (if unacceptable)

oversight, at worst this could be construed as deliberate attempt to mislead the reader. The details of these, and other issues, are outlined below.

1. The statement in the Abstract “here we report the identification of a novel class of chromatin remodeling enzyme,” is simply untrue. This manuscript does not “identify” a novel chromatin regulator. A role for these type of AAA+ ATPases in the regulation of chromatin is well established in multiple systems.
2. Another example is the degree to which the study by Cho et. al., 2019 (published in this journal) is marginalized. This study reported the first cryo-EM structure of a ANCCA/ATAD2 type AAA+ ATPases (in ADP, ATP and apo states) and also demonstrated histone deposition function in vitro. This study provides a structural basis for understanding the molecular mechanisms of these enzymes and deserves proper recognition and acknowledgement here. Not just a passing sentence in the introduction suggesting that it “might play a role in nucleosome assembly”
3. The statement, “so far, no segregase with chromatin remodeling activity has been described in any organism” is disingenuous. By “segregase with chromatin remodeling activity” the authors mean the ability to evict histones from chromatin which is precisely the function that has been proposed for Yta7 and supported by in vivo analyses. (Lombardi et al 2011 and 2015).
4. The statement in the introduction “here, we identify the AAA+-ATPase containing Yta7 protein from *Saccharomyces cerevisiae* as a novel factor required for chromatin replication in vivo and in vitro” is not accurate. Required means indispensable or essential and Yta7 is not essential for replication in the in vitro system. Furthermore, *yta7Δ* and other *yta7* mutants have only a very modest S phase delay at best, so they are certainly not essential in vivo. The FACS data presented in Fig 1C are not convincing. The number of repeats should be indicated and the difference in the percentage of cells in S phase relative to wild type should be estimated and shown to be statistically significant. The number of repeats for the assays is missing from many of the figures and should be included.
5. The statement “the molecular consequences of these phosphorylation events, however, remain unknown” is particularly puzzling given that the corresponding author has previously published a paper almost a decade ago that addressed the function of these phosphorylation sites (Kurat et al 2011). In fact, a conclusion from the 2011 study is that CDK-dependent phosphorylation of Yta7 drives its release from chromatin. In this manuscript the conclusion is that phosphorylation controls the ability of Yta7 to evict histones but chromatin recruitment/binding is not affected by phosphorylation. This apparent contradiction needs to be properly acknowledged and addressed. Information giving the positions of the phosphorylation sites should be included in this manuscript so that the reader does not have to back through the literature for these details. With regard to these sites are there actually any data which shows that show that any of these sites are actually phosphorylated in vivo? If so which ones? What impact does mutation of the CK2 phosphorylation sites have on activity in these assays? The ability of Yta7 purified from cells outside of G1 to evict histones should be investigated.
6. Analysis of Yta7 genetic interactions is used to suggest that “YTA7 may also have a role in DNA synthesis”. It should be acknowledged at this point in the manuscript that there is already evidence that indicates a role for Yta7 and its homologs in replication. For example, both Yta7 and human ANCCA/ATAD2, have been shown to co-purify with replication proteins, ANCCA/ATAD2 is known to be recruited to replication sites through a direct interaction with acetylated histones. Furthermore, a conserved interaction between Yta7 type proteins and the FACT histone chaperone which processes histones during replication have been noted.
7. Figure 2A and the paragraph at the top of page 7 are not necessary. The relationship between Yta7 and its counterparts in other organisms to segregases such as p97/Cdc48 and NSF have been documented. In fact, Cho et al 2017 have compared the structure of *S. pombe* Abo1 with p97/Cdc48 and NSF segregases. It should be noted in the manuscript that the findings reported in this study for Yta7 (Fig 2B and C) are consistent with those previously reported by Cho et al for Abo1 (ie six-fold symmetry and ATP binding unnecessary for hexamer formation).
8. The validity of many of the experiments relies on the purity of the protein preps. For example, it is clear that the Yta7 preparations contain multiple bands. What are these? Have these preparations been checked for the presence of other significant proteins e.g. FACT subunits, Nhp6 etc.? Can the authors exclude the possibility that differences in activity of the wild type and mutants are due to differences in co-purifying proteins? Similarly, in Fig 4AB can phosphorylation of Yta7 by a contaminating kinase be excluded. Incorporation of a control showing that phosphorylation is blocked by the addition of Sic1 would help here.
9. Does the phospo mutant form hexamers? These data are missing from Fig S2C
10. With respect to the statement in the discussion “Yta7’s non-canonical bromodomain was

shown to bind histones independently of posttranslational modifications (Gradolatto et al 2009)“ It is probably worth noting in the discussion that the same study concluded that “regions of Yta7 other than the bromodomain conferred histone association”

Reviewer #3 (Remarks to the Author):

The manuscript by Kurat and colleagues describes the genetic and biochemical analysis of the function of the Yta7 protein. In particular, the authors focus on the potential role of this protein in regulating chromatin structure during DNA replication. The authors show that a YTA7 deletion, mutation of Yta7 CDK phosphorylation sites, or a Yta7 ATP-binding motif mutation each result in a mild S phase progression defect. They go on to show that Yta7 forms a typical hexameric AAA+ ATPase structure and that it binds acetylated nucleosomal DNA. Consistent with an S-phase function, they find that modification of Yta7 with CDK strongly activates the Yta7 ATPase. Finally, they show that Yta7 removes histones from nucleosomal DNA in an ATP- and S-CDK-dependent manner and that addition of Yta7 to in vitro chromatin replication assays enhances the amount of replication products.

The strength of this paper are the biochemical studies addressing the interaction of Yta7 with nucleosomal DNA and its ability to remove histones from nucleosomal DNA. These findings show that Yta7 specifically binds to nucleosomal DNA and that this is significantly activated by prior histone acetylation (via pSAGA). In addition, the authors provide strong evidence that Yta7 ATPase activity is activated by prior S-CDK phosphorylation. Finally, they find that an important consequence of Yta7 action is the removal of histones from nucleosomal DNA resulting in lower nucleosome density. Together these data provide important new information about Yta7 function and how it can be regulated during the cell cycle.

The weaker part of this manuscript is the evidence that Yta7 functions at replication forks. The genetic evidence that Yta7 is directly involved in replication is primarily at the level of previous global genetic interaction studies and a weak effect on progression through S phase. Each of these observations could be explained by indirect effects of other known functions of Yta7 (e.g. regulation of histone transcription or global consequences of altered nucleosome patterns). Much of the evidence that Yta7 is dedicated to replication relies on the in vitro replication assays shown. Unfortunately, there is nothing in these assays to suggest that this effect is specific to replication. A simple explanation of the data is that reduced nucleosome density leads to more efficient replication, something that is already known (e.g. compare Fig. 5E to Supp. Fig 3E). The authors do not address whether Yta7 works better than any of the other nucleosome remodeling complexes that they previously showed enhance replication. For example, if Yta7 were dedicated to replication, one would expect that it functions better than = other nucleosome remodeling complexes that have previously been shown to enhance replication (e.g. INO80 and ISW1a). For example, there are no experiments in which Yta7 is added in addition to these remodeling complexes or comparing the effectiveness of Yta7 to these other activities. More importantly, given that the authors show convincingly that Yta7 is recruited by histone acetylation, it is not clear that Yta7 is normally recruited to replication forks. Does Yta7 have any effect on replication when the nucleosomes are not acetylated? Also, if Yta7 is acting at the replication fork one would expect for it to be detected iPond or similar experiments. Is this the case?

Thus, while this is a very nice analysis of the function of Yta7 on nucleosomal templates, the evidence that Yta7 is a replication-specific nucleosome segregase is lacking. A manuscript with additional evidence supporting this hypothesis or re-written to eliminate the replication claims would be a stronger candidate.

Specific point:

Because the mutation in the ATP binding and hydrolysis domain is predicted to prevent ATP binding and hydrolysis, it would be better to refer to it as an “ATP-binding” rather than an “ATPase” mutation. An ATPase mutant would be much more likely to be generated by a mutation in the R-finger of Yta7 (although it would be important to confirm this).

REVIEWER COMMENTS

Firstly, we would like to thank all reviewers for their constructive evaluations of our manuscript. We addressed all reviewers comments. To help navigate through the revisions, we highlighted all changes in red within the text.

Below, please find the detailed responses to the reviewers' comments:

Reviewer #1 (Remarks to the Author):

This paper shows that budding yeast lacking the AAA+ATPase protein Yta7 have a minor (5 minute) delay in progression through S-phase, and that they have more DNA damage and more recombination (Fig. 1). Yta7 sequence is similar to that of segregases and is a hexamer (Fig. 2). In vitro, Yta7 binds to chromatin rather than naked DNA, and this is enhanced by histone acetylation (Fig. 3). In vitro, Yta7 can be phosphorylated by CDK from S phase cells and this increases its ATPase activity in vitro (Fig. 4). Yta7 from S phase cells can displace histones from reconstituted chromatin and can promote replication of chromatinized templates in vitro (Fig. 5). The conclusion from these data is made that the Yta7 is a chromatin segregase promoting chromatin replication. This is an interesting idea, but needs more experimental data to support that this is the physiological role of Yta7 rather than an in vitro artifact coupled with the in vivo consequences of the known increased histone expression that occurs in yeast lacking Yta7.

We wish to thank the reviewer for her/his constructive comments, which certainly helped improve the paper. We are happy to see that the reviewer finds our model of how Yta7 acts as a chromatin segregase to support chromosome replication an “*interesting idea*”. Indeed, we believe that the strength of this paper are the biochemical data, showing in detail how a chromatin modifying enzyme can be regulated by cell cycle kinases. However, we agree that more *in vivo* data on the physiological role of Yta7 would make the paper stronger and would corroborate our claim that Yta7 influences chromosome replication. We now included a substantial amount of new data, which helped to support our conclusions.

Major concerns:

1. There is a large body of literature that shows that Yta7 is required to repress histone expression during phases of the cell cycle other than S. As a consequence, without Yta7, there is excess histones in the cells and excess histones on a chromatin. These facts are not even mentioned in this paper! The excess histones in the Yta7 could be the indirect explanation for the slower S-phase and more DNA damage, given that the Gunjan / Verreault labs have a large body of publications showing that overexpression of histones have these effects.

Yta7 is involved in both major S phase events, histone gene transcription and, as we now show, chromosome replication. It is therefore very difficult to dissect to which extent both events contribute to our *in vivo* phenotypes by just using cell-based assays. We think that our approach, using *in vivo* assays (Figure 1) as well as by directly showing that Yta7 influences chromatin replication in an *in vitro* system is quite powerful (Figure 5). It is true that overexpression of histones

can lead to similar *in vivo* phenotypes we observed in this manuscript. Importantly and in contrast what the reviewer was pointing out, elimination of Yta7 function does not lead to overexpression of histone genes outside of S phase and subsequent overrepresentation of soluble histones (Fillingham et al., 2009; Kurat et al., 2011). At histone gene promoters, elimination of Yta7 leads to increased chromatin density (Kurat et al., 2011; Lombardy et al., 2011), generating a repressive chromatin structure to hinder proper function of the transcription machinery, thus, reducing histone gene transcription.

At the time, we were proposing that Yta7 might act as boundary element to prevent spreading of repressive chromatin by keeping histone chaperone Rtt106 and chromatin remodeler RSC in place (Fillingham et al., 2009; Kurat et al., 2011; reviewed in Kurat et al., 2014). Others were proposing another interesting idea that Yta7 might be acting directly to disassemble nucleosomes, to support DNA template processes like histone gene transcription (Lombardi et al., 2011; Cattaneo et al., 2014). Importantly, in our paper, we now biochemically confirmed this hypothesis and uncovered a unique mechanism regulating this process and further showed that Yta7 influences DNA replication. We now included these important models in the introduction now.

As stated by the reviewer, the reduction of histone gene transcription by the loss of Yta7's function might contribute to observed *in vivo* phenotypes and we now discussed this possibility.

However, because

- 1) of the new *in vivo* data we are now including (see point 2)
- 2) Yta7 strongly supports chromatin replication in our purified reconstituted system (which excludes indirect effects by e.g. transcription)
- 3) low levels of Yta7 (25 nM) compared to the chromatin template (200 nM nucleosomes) have proven sufficient for the enhancement of chromatin replication *in vitro*, consistent with a high local concentration residing at the replication fork.
- 4) Yta7's homologue ATAD2 localizes to sites of active replication (Koo S.J. et al., 2016),

we are confident that our *in vivo* replication phenotypes are rather direct and not caused by indirect effects.

2. There is no demonstration that Yta7 localizes to chromatin in S phase *in vivo* or to sites of DNA replication, which would be critical to rule out such an indirect role for Yta7's influence on replication and genome stability.

We totally agree with the reviewer on that point. We now provide two lines of evidence supporting our claim that Yta7 is involved in chromosome replication. Firstly, by using ChIP following qPCR, we show that Yta7 clearly localizes to a selection of important origins (including early and late origins) of replication in S phase (**new Supplementary Fig. 2b**). On top of the reviewer's request, we observed physical interaction with replication-specific proteins. Using *in vivo* pull-down assays, we show that Yta7 physically interacts with the origin recognition complex (ORC) as well as the replicative helicase MCM (**new Supplementary Fig. 2a**). These data provide additional support and strengthen our claim that Yta7 influences S phase chromosome replication directly.

3. Yta7 has previously been proposed to be a histone disassembly factor and this should be discussed (discussed in the review by Cattaneo et al., *Molecules and Cells* 2014).

As discussed in point 1, we extended our introduction and discussed this more thoroughly. We also included the reference in this paragraph (we had it included already at another part of the paper).

4. Others have published that in mid S phase, Yta7 is heavily phosphorylated by Cdk1 and CK2, which is required for Yta7 dissociation from histone gene promoters. This should be discussed. Is this the same phosphorylation that they are seeing here by CDK – what do they even mean by CDK here actually?

We have previously shown that Yta7 is a target of CDK and CK2 *in vivo* (Kurat *et al.*, 2011). We mutated both CDK as well as CK2 phosphorylation sites separately and the observed phenotypes, e.g. reduction of histone gene transcription, were quite clear for CDK mutants. In the case of CK2 mutants, reduction of histone gene transcription was extremely mild. The role for CK2 is not clear at the moment. Our hypothesis is now that the main function of Yta7 is during S phase to support the major S phase events, histone gene transcription and chromosome replication and is regulated by S phase forms of CDK (S-CDK, Clb5-Cdc28-Csk1). In our previous work, we could not directly show that Yta7 is a S-CDK target, because we had no purified S-CDK available. The Diffley lab then invested quite some effort to generate an expression strain bearing a non-degradable form of the S phase cyclin Clb5. Using this strategy it was possible to purify S-CDK (Fig. 4a) to directly study Yta7 S phase phosphorylation and consequences in detail. The phosphorylation mutant we use in our paper is the CDK mutant lacking the phosphor-sites described in our previous work (Kurat *et al.*, 2011). We clarified this in the text.

However, others have shown that Yta7 has roles beyond S phase as well (Lombardi *et al.*, 2011). We hypothesize now in our paper that, outside S phase, Yta7 might be regulated by either G2 or M phase forms of CDK and/or by CK2. This is an intriguing idea and we currently have a project to dissect non-S phase functions of Yta7.

5. The kinase reactions need to include more controls. Firstly another protein with serines that is not normally phosphorylated by S-CDK, to show they are not just adding excess kinase that would force the phosphorylation of anything. To conclude it is specific to S-phase CDK as they do, they need to show that CDK from other phases of the cell cycle does not have this effect.

We agree with the reviewer that *in vitro* kinase assays can lead to unspecific phosphorylation events, especially by adding excess of substrate and/or kinase. We therefore used low amounts of kinase and substrate in all our assays (25 nM). To address the reviewers concerns, we included three new experiments (new Supplementary Fig. 5):

1. We included a control where we added increasing amounts of the S-CDK inhibitor Sic1. Addition of Sic1 clearly abolishes S-CDK phosphorylation of Yta7 (new Supplementary Fig. 5a), indicating that Yta7 is a direct target of S-CDK and not of a contaminating kinase.

2. We included a positive control using the histone chaperone CAF-1 as a substrate. The Cac1 subunit of CAF-1 is a known S-CDK target (Jeffrey et al., 2015). As expected, we show that Cac1 is clearly phosphorylated by S-CDK, whereas the other subunits of the complex (Cac2 and Msi1), although bearing serines and threonines, were not (new Supplementary Fig. 5 b and c).
3. Additionally and as requested by the reviewer, we included a negative control. FACT, a histone chaperone involved in S phase chromatin replication and not described as a S-CDK target was not phosphorylated by S-CDK in our assay. Importantly, FACT bears many serines/threonines, some part of a CDK consensus sequence, in its primary sequence (new Supplementary Fig. 5 b and c).

Together, we feel that these new experiments now show that our *in vitro* assay is very specific and corroborate our claim that Yta7 is a direct S-CDK target. As discussed in point 4, we propose that Yta7's main function is to support S phase-specific events like genome replication. Yta7's role outside of S phase, which might involve phosphorylation by non-S phase-forms of CDK and/or CK2, is interesting and an ongoing project and we therefore feel that this is out of the scope of this manuscript.

6. To support their conclusions, they need to show that Yta7 does not get phosphorylated in S phase upon inactivation of CDK *in vivo*.

We have demonstrated this previously (Kurat et al., 2011). *In vivo*, we observed a robust phosphatase-sensitive mobility shift of Yta7 in S phase. This phospho-shift was absent in a S-CDK deficient strain lacking the S phase cyclins Clb5 and Clb6 or by inhibiting CDK.

7. For the histone removal experiments in Fig. 5 that lead to replication, they need to show that this is unique to Yta7 and not to other chromatin remodelers or histone chaperones, because such experiments are prone to biochemical artifacts. Add enough of any histone binding protein / molecule and it will pull histones off of the DNA. What is the stoichiometry of Yta7 to nucleosomes in these experiments?

We agree that this is an important control. We now included new experiments where we tested if the histone chaperone FACT/Nhp6 or chromatin remodeler ISW1a can act similarly as Yta7 plus and minus kinase. As with experiments involving Yta7, we measured the histone/DNA ratio and could not observe any chromatin segregase activity (new Supplementary Fig. 6). This shows that Yta7's function seems to be unique and that other important chromatin factors involved in chromatin replication behave differently. As with replication experiments, low amounts of Yta7 (25 nM) were incubated with chromatin templates (200 nM nucleosomes).

Other concerns:

1. The chromatin templates in Fig. 3B have hugely varying ratios of DNA to H3 to H2A. These experiments need to be made more consistent.

We added a new experiment showing H2A and H3 blots with more consistent DNA/histone ratios (new Fig. 3B).

Reviewer #2 (Remarks to the Author):

This study demonstrates uses predominantly in vitro approaches to show that the *S. cerevisiae* Yta7 AAA+ ATPase is able evict histones from chromatin in vitro and that the activity of this enzyme is controlled by phosphorylation. These are significant findings that provides important insight to function of this class of chromatin remodeler and given the role of its human homolog (ANCCA/ATAD2) in cancer will be of general interest.

There are however, important issues that need to be addressed and the manuscript needs extensive modification. A major problem is that some of conclusions/claims are exaggerated and not supported by the data. Another problem is the degree to which data from previous studies from other groups are marginalized or ignored. At best this is an inadvertent (if unacceptable) oversight, at worst this could be construed as deliberate attempt to mislead the reader. The details of these, and other issues, are outlined below.

We are very thankful for the reviewer's constructive comments, which certainly helped improve the paper. We are happy to see that the reviewer feels that our findings "*provide important insights*" into Yta7's functions and that our results will be of "*general interest*". However, we are also grateful that the reviewer points out that the manuscript will improve by tuning down some claims and by including more discussion of previous research. This was an honest oversight from our side and we apologize for this. Below, please find the detailed responses to the comments:

1. The statement in the Abstract "here we report the identification of a novel class of chromatin remodeling enzyme," is simply untrue. This manuscript does not "identify" a novel chromatin regulator. A role for these type of AAA+ ATPases in the regulation of chromatin is well established in multiple systems.

We agree with the reviewer on that point. It is true that there are some reports about this emerging class of chromatin remodeling function. However, these studies mostly used *in vivo*, cell-based assays to corroborate their claims. These are valid and important approaches, however, biochemical data were missing and it was therefore never possible to directly prove the models and to rule out indirect effects. Recent structural work by Cho et al., 2019 (also mentioned in point 2), made important contributions towards our understanding of this emerging class of chromatin modifying machines. We feel that AAA⁺-ATPases involved in chromatin replication, although described previously, are still poorly understood.

However, we agree with the reviewer that we overstated this in the abstract and changed that into "*Here, we report the characterization of a chromatin remodeling enzyme, entirely distinct from classical SNF2-ATPase family remodelers*".

2. Another example is the degree to which the study by Cho et. al., 2019 (published in this journal) is marginalized. This study reported the first cryo-EM structure of a ANCCA/ATAD2 type AAA+ ATPases

(in ADP, ATP and apo states) and also demonstrated histone deposition function in vitro. This study provides a structural basis for understanding the molecular mechanisms of these enzymes and deserves proper recognition and acknowledgement here. Not just a passing sentence in the introduction suggesting that it “might play a role in nucleosome assembly”

We agree with the reviewer and apologize for this oversight. Indeed, the Cho et al., 2019 study makes important contributions towards our understanding of this class of AAA⁺-ATPases by resolving most of the structure of Abo1, fission yeasts homologue of Yta7.

For their claim that Abo1 might be an ATP hydrolysis-dependent histone chaperone, the authors used single molecule/DNA curtain approaches to study histone deposition. Interestingly, micrococcal nuclease protection assays with Abo1’s assembly products revealed the presence of quite different DNA fragments as normally seen by histone chaperones like FACT or CAF-1. The nature of these intermediates is not quite clear at the moment. It might be another chaperone involved in this, or it might also be that histones need to be acetylated in their assays. In line with this, we now provide new data showing that Yta7 does not support chromatin replication when chromatin templates were not acetylated (new Supplementary Fig. 7). We agree with the reviewer that Abo1 needs to be more acknowledged and we extended the discussion.

3. The statement, “so far, no segregase with chromatin remodeling activity has been described in any organism” is disingenuous. By “segregase with chromatin remodeling activity” the authors mean the ability to evict histones from chromatin which is precisely the function that has been proposed for Yta7 and supported by in vivo analyses. (Lombardi et al 2011 and 2015).

We agree and changed the text accordingly.

4. The statement in the introduction “here, we identify the AAA⁺-ATPase containing Yta7 protein from *Saccharomyces cerevisiae* as a novel factor required for chromatin replication in vivo and in vitro” is not accurate. Required means indispensable or essential and Yta7 is not essential for replication in the in vitro system. Furthermore, yta7Δ and other yta7 mutants have only a very modest S phase delay at best, so they are certainly not essential in vivo. The FACS data presented in Fig 1C are not convincing. The number of repeats should be indicated and the difference in the percentage of cells in S phase relative to wild type should be estimated and shown to be statistically significant. The number of repeats for the assays is missing from many of the figures and should be included.

Again, we agree with the reviewer and changed *required* to *promoting*. One big problem with *in vivo* approaches to study chromatin-modifying enzymes is their redundancy. We were quite surprised to see any defect in S phase at all. Because we now know from our *in vitro* studies, that ATPase is regulated by phosphorylation and that both functions are essential for the replication phenotype, the use of three different yta7 mutations (Δ, ATP-binding and Phospho) in Fig. 1 can be seen as biological replicates already. However, we repeated the experiment three times and we now include this information in the Figure legend.

5. The statement “the molecular consequences of these phosphorylation events, however, remain unknown” is particularly puzzling given that the corresponding author has previously published a

paper almost a decade ago that addressed the function of these phosphorylation sites (Kurat et al 2011). In fact, a conclusion from the 2011 study is that CDK-dependent phosphorylation of Yta7 drives its release from chromatin. In this manuscript the conclusion is that phosphorylation controls the ability of Yta7 to evict histones but chromatin recruitment/binding is not affected by phosphorylation. This apparent contradiction needs to be properly acknowledged and addressed.

These are important points raised by the reviewer. In our paper in 2011, we observed S phase histone transcription phenotypes *in vivo*. However and in contrast what the reviewer was stating, at this time we had no idea about the molecular function of Yta7. At that time, we were proposing that Yta7 might act as boundary element to prevent spreading of repressive chromatin by keeping histone chaperone Rtt106 and chromatin remodeler RSC in place (Fillingham et al., 2009; Kurat et al., 2011; Kurat et al., 2014). Others were proposing another, exciting idea that Yta7 might be directly involved in the disassembly of nucleosomes (see also point 3, Lombardi et al., 2011; Cattaneo et al., 2014). Importantly this could never be confirmed and with our paper, we now biochemically confirmed this hypothesis. On top of that, we uncovered a unique mechanism regulating this process and further linked Yta7 to chromosome replication. As requested, we have now discussed this in the introduction.

Information giving the positions of the phosphorylation sites should be included in this manuscript so that the reader does not have to back through the literature for these details. With regard to these sites are there actually any data which shows that show that any of these sites are actually phosphorylated *in vivo*?

If so which ones? What impact does mutation of the CK2 phosphorylation sites have on activity in these assays? The ability of Yta7 purified from cells outside of G1 to evict histones should be investigated.

We added information about the CDK phosphorylation sites. We and others found these sites to be phosphorylated *in vivo* (discussed in Kurat et al., 2011). Regarding CK2, we have previously shown that Yta7 is a target of not only CDK but also CK2 (Kurat *et al.*, 2011). While mutations of the CDK sites resulted in clear phenotypes in regard of S phase histone gene transcription, the role for CK2 was not clear. However, others have shown that Yta7 has roles beyond S phase as well (Lombardi et al., 2011). We hypothesize now in our paper that, outside S phase, Yta7 might be regulated by either G2 or M phase forms of CDK and/or by CK2 to support e.g. transcription. This is an intriguing idea and we currently have a project to dissect non-S phase functions of Yta7, which is out of the scope of this manuscript.

6. Analysis of Yta7 genetic interactions is used to suggest that “YTA7 may also have a role in DNA synthesis”. It should be acknowledged at this point in the manuscript that there is already evidence that indicates a role for Yta7 and its homologs in replication. For example, both Yta7 and human ANCCA/ATAD2, have been shown to co-purify with replication proteins, ANCCA/ATAD2 is known to be recruited to replication sites through a direct interaction with acetylated histones. Furthermore, a conserved interaction between Yta7 type proteins and the FACT histone chaperone which processes histones during replication have been noted.

We acknowledged these physical interactions with the replisome, FACT as well as the recruitment of ATAD2 to replication proteins already in the original manuscript. However, to strengthen our point that Yta7 is involved in chromosome replication *in vivo*, we included more data now:

1. By using ChIP following qPCR, we show that Yta7 clearly localizes to a selection of important origins of replication in S phase (new Supplementary Fig.2b).
2. We observed physical interaction of Yta7 with replication-specific proteins. Using pull-down assays, we show that Yta7 physically interacts with the origin recognition complex (ORC) as well as the replicative helicase MCM (new Supplementary Fig. 2a).

7. Figure 2A and the paragraph at the top of page 7 are not necessary. The relationship between Yta7 and its counterparts in other organisms to segregases such as p97/Cdc48 and NSF have been documented. In fact, Cho et al 2017 have compared the structure of *S. pombe* Abo1 with p97/Cdc48 and NSF segregases. It should be noted in the manuscript that the findings reported in this study for Yta7 (Fig 2B and C) are consistent with those previously reported by Cho et al for Abo1 (ie six-fold symmetry and ATP binding unnecessary for hexamer formation).

We agree with the reviewer and omitted these panels and the paragraph and included the reference.

8. The validity of many of the experiments relies on the purity of the protein preps. For example, it is clear that the Yta7 preparations contain multiple bands. What are these? Have these preparations been checked for the presence of other significant proteins e.g. FACT subunits, Nhp6 etc.? Can the authors exclude the possibility that differences in activity of the wild type and mutants are due to differences in co-purifying proteins? Similarly, in Fig 4AB can phosphorylation of Yta7 by a contaminating kinase be excluded. Incorporation of a control showing that phosphorylation is blocked by the addition of Sic1 would help here.

We feel that the quality of our Yta7 preps are high. However, the "multiple bands" the reviewer is referring to, are break-down products of Yta7 (as determined by immune-blotting, data not included). As Yta7 alone shows no activity without S-CDK phosphorylation, it is highly unlikely already that this might be due to contaminating chromatin factors (Fig. 5b). Importantly, all necessary controls (ATP-binding-, Phospho-mutants as well as S-CDK alone), did not show activity (Fig. 5c). This was not due to mis-localization of the Yta7 mutant proteins to chromatin (Supplementary Fig. 4b), or because of a problem in the formation of the hexameric structure (Supplementary Fig. 3b, new Figure contains also requested Phospho-mutant SEC profile, see also point 9). All these results clearly exclude the possibility that our results are caused by contaminating factors. However, to further corroborate our findings, we included an additional control, showing that Yta7's chromatin segregase activity is different to other chromatin factors. We included new data showing that the histone chaperone FACT/Nhp6 or the chromatin remodeller ISW1a cannot disassemble chromatin, even in the presence of S-CDK (new Supplementary Fig. 6).

As requested, for our *in vitro* kinase reaction, we added the Sic1 control (Supplementary Fig. 5a). Our data clearly show that Yta7 phosphorylation by S-CDK is abolished by Sic1 addition, showing that the reactions are specific to S-CDK and no contaminating kinase.

9. Does the phospho mutant form hexamers? These data are missing from Fig S2C

As mentioned in point 8, we included this experiment. New Supplementary Fig. 3b clearly shows that Yta7 phospho-mutant form hexamers.

10. With respect to the statement in the discussion “Yta7’s non-canonical bromodomain was shown to bind histones independently of posttranslational modifications (Gradolatto et al 2009)” It is probably worth noting in the discussion that the same study concluded that “regions of Yta7 other than the bromodomain conferred histone association”

As suggested, we added this information.

Reviewer #3 (Remarks to the Author):

The manuscript by Kurat and colleagues describes the genetic and biochemical analysis of the function of the Yta7 protein. In particular, the authors focus on the potential role of this protein in regulating chromatin structure during DNA replication. The authors show that a YTA7 deletion, mutation of Yta7 CDK phosphorylation sites, or a Yta7 ATP-binding motif mutation each result in a mild S phase progression defect. They go on to show that Yta7 forms a typical hexameric AAA+ ATPase structure and that it binds acetylated nucleosomal DNA. Consistent with an S-phase function, they find that modification of Yta7 with CDK strongly activates the Yta7 ATPase. Finally, they show that Yta7 removes histones from nucleosomal DNA in an ATP- and S-CDK-dependent manner and that addition of Yta7 to *in vitro* chromatin replication assays enhances the amount of replication products.

We are grateful for the reviewer’s positive and constructive comments. We are very happy to see that the reviewer feels our study provides “*important new information about Yta7 function and how it can be regulated during the cell cycle*” and that it is “*a very nice analysis of the function of Yta7 on nucleosomal templates*”.

The strength of this paper are the biochemical studies addressing the interaction of Yta7 with nucleosomal DNA and its ability to remove histones from nucleosomal DNA. These findings show that Yta7 specifically binds to nucleosomal DNA and that this is significantly activated by prior histone acetylation (via pSAGA). In addition, the authors provide strong evidence that Yta7 ATPase activity is activated by prior S-CDK phosphorylation. Finally, they find that an important consequence of Yta7 action is the removal of histones from nucleosomal DNA resulting in lower nucleosome

density. Together these data provide important new information about Yta7 function and how it can be regulated during the cell cycle.

We agree, we believe that the real strength of this paper are the biochemical data, showing in detail how a chromatin modifying enzyme can be regulated by cell cycle kinases.

The weaker part of this manuscript is the evidence that Yta7 functions at replication forks. The genetic evidence that Yta7 is directly involved in replication is primarily at the level of previous global genetic interaction studies and a weak effect on progression through S phase. Each of these observations could be explained by indirect effects of other known functions of Yta7 (e.g. regulation of histone transcription or global consequences of altered nucleosome patterns).

We agree again that the weaker part of the manuscript are the genetic cell-based assays.

Much of the evidence that Yta7 is dedicated to replication relies on the *in vitro* replication assays shown. Unfortunately, there is nothing in these assays to suggest that this effect is specific to replication. A simple explanation of the data is that reduced nucleosome density leads to more efficient replication, something that is already known (e.g. compare Fig. 5E to Supp. Fig 3E). The authors do not address whether Yta7 works better than any of the other nucleosome remodeling complexes that they previously showed enhance replication. For example, if Yta7 were dedicated to replication, one would expect that it functions better than = other nucleosome remodeling complexes that have previously been shown to enhance replication (e.g. INO80 and ISW1a). For example, there are no experiments in which Yta7 is added in addition to these remodeling complexes or comparing the effectiveness of Yta7 to these other activities.

We understand the reviewer's concern. However, low levels of Yta7 (25 nM) compared to the chromatin template (around 200 nM nucleosomes) have proven sufficient for the enhancement of chromatin replication (Fig. 5e), consistent with a high local concentration residing at the replication fork. Furthermore, we incubate the replication reactions for only 5 minutes, where the bulk of chromatin in the assay is still unchanged (Fig. 5b). This suggests that enhancement of chromatin replication by Yta7 is likely to be caused by the disassembly of nucleosomes in close proximity to the replisome, not by merely generating "naked" DNA stretches.

But we do agree with the reviewer that, standing on its own, our *in vitro* assay is not the "golden bullet", but neither are *in vivo* assays. So we believe that a combination of both, *in vitro* and *in vivo* is quite powerful in identifying novel factors in chromosome replication, like Yta7.

How Yta7 compares to other chromatin remodelers, or better, how different remodeling activities operate together to support chromatin replication is an important and very exciting question. Our hypothesis now is that different remodelers (like INO80, ISW1a, Yta7,...) might be acting on different origins at different sites of the genome at different times during S phase. We have a current project, which is exactly trying to answer these questions. Trying different combinations of remodelers like INO80 and ISW1a how they compare to Yta7 activity is interesting, but we feel these experiments would not add much to the story at this point. We also feel that this would be out of the scope of the manuscript.

More importantly, given that the authors show convincingly that Yta7 is recruited by histone acetylation, it is not clear that Yta7 is normally recruited to replication forks. Does Yta7 have any effect on replication when the nucleosomes are not acetylated? Also, if Yta7 is acting at the replication fork one would expect for it to be detected iPond or similar experiments. Is this the case?

Here we completely agree with the reviewer. Showing that Yta7 is actually present at replication origins is important and would strengthen our *in vivo* data and corroborate our claim that Yta7 is promoting replication directly. As suggested, we were using our *in vitro* replication assay, to determine if Yta7 promotes chromatin replication when histones were not acetylated. Our **new Supplementary Fig. 7** now shows that Yta7 does not support chromatin replication under these conditions, which is in-line with our recruitment experiments of Yta7 to acetylated chromatin (Fig. 3b). This also nicely fits with previous *in vivo* experiments showing that replication origins are highly acetylated at early S phase, suggesting a mechanism to recruit Yta7 to replication origins *in vivo* (Unnikrishnan et al., 2010).

To further strengthen our *in vivo* experiments, we included two more experiments:

1. By using ChIP following qPCR, we now show that Yta7 clearly localizes to a selection of important origins of replication in S phase (including early and late origins) (**new Supplementary Fig.2b**).
2. We observed physical interaction of Yta7 with replication-specific proteins. Using pull-down assays, we show that Yta7 physically interacts with the origin recognition complex (ORC) as well as the replicative helicase Mcm2-7 (**new Supplementary Fig. 2a**).

Thus, while this is a very nice analysis of the function of Yta7 on nucleosomal templates, the evidence that Yta7 is a replication-specific nucleosome segregase is lacking. A manuscript with additional evidence supporting this hypothesis or re-written to eliminate the replication claims would be a stronger candidate.

We completely agree.

Because:

- 1) of the new *in vitro* and *in vivo* data we are now including
- 2) Yta7 strongly supports chromatin replication in our purified reconstituted system (which excluded indirect effects by e.g. transcription)
- 3) low levels of Yta7 (25 nM) compare to the chromatin template (around 200 nM nucleosomes) have proven sufficient for the enhancement of chromatin replication *in vitro*, consistent with a high local concentration residing at the replication fork
- 4) Genetic interaction data
- 5) Yta7's homologue ATAD2 localizes to sites of active replication (Koo S.J. et al., 2016)

we are confident that our Yta7 is indeed a replication-specific nucleosome segregase.

Specific point:

Because the mutation in the ATP binding and hydrolysis domain is predicted to prevent ATP binding and hydrolysis, it would be better to refer to it as an “ATP-binding” rather than an “ATPase” mutation. An ATPase mutant would be much more likely to be generated by a mutation in the R-finger of Yta7 (although it would be important to confirm this).

We agree and changed that throughout the manuscript.

REVIEWER COMMENTS

Reviewer #1 (Remarks to the Author):

The authors have done a better job at more accurately describing their work in the context of the literature now.

Other problems remain - I am underwhelmed by the new experiments to support a direct role in replication in cells, i.e. the new ChIP and IP analyses.

They show 1 primer pair for a bunch of different origins to which it binds to all of them. However, they have to show some specificity of the binding by including primer pairs for regions that it doesn't bind to, as it could just be binding all over to the chromatin. Also a temporal analysis showing it binds specifically in S phase would have been useful. Once a rigorous ChIP analysis is performed, it needs to be placed in the main figure not the supplemental.

The IP shows that FLAG tagged Yta7 brings down some replication proteins. However, given that Yta7 binds to chromatin, they need to either add a nuclease or EtBr to show that the interaction is not through the DNA, as otherwise they are just pulling down chromatin and every protein on chromatin. Again, doing this analysis in a temporal manner would have been useful, and they must also blot for chromatin proteins that it does not bind too, to establish specificity. Once rigorous IPs are completed, they should be included in the main figure not the supplemental.

Regardless, even if they could show that Yta7 goes to replication fork, it seems more likely to be involved in events after DNA replication given that it only is recruited to acetylated chromatin and the literature on the human homolog.

They also cite the fact that human counterpart of Yta7, ATAD2, binds to replication sites in support of Yta7 doing the same. But ATAD2 specifically binds to the histone modifications that are found on newly-synthesized histones (H4K5 and 12) (Kool et al nature communications 2016) and on H3 (Revenko 2010), which would position it on the newly-replicated chromatin, not for a role in disassembling chromatin. This paper also shows that ATAD2 depletion causes a replication defect which is what you see when you get rid of factors that assemble new histones into chromatin. i.e. Stillman and others have shown that depletion of proteins that assemble newly-synthesized H3/H4 i.e. CAF-1 feedback to block replication, which could be what is happening here with ATAD2, and potentially also Yta7.

The fact that Yta7 only binds to acetylated chromatin that have H3 N-terminal acetylations (same as the ones that ATAD2 binds to) indicates that it is likely doing the same thing as ATAD2, i.e. after replication in vivo. The biochemical disassembly assay is likely to be an in vitro artifact. As such there is no evidence that Yta7 is disassembling chromatin in vivo.

Reviewer #2 (Remarks to the Author):

The authors have satisfactorily addressed the issues raised by this reviewer. The modifications to the text and inclusion of the additional data have strengthened the manuscript significantly.

One very minor point. As far as I understand it, Drosophila does not have an ATAD2/Yta7 homolog so perhaps the text in the final paragraph page 4 should read, "Yta7 is conserved among eukaryotes....." rather than "conserved among all eukaryotes".

Reviewer #3 (Remarks to the Author):

The revised manuscript by Kurat and colleagues addresses the role of the Yta7 ATPase in chromosomal replication. They have identified an intriguing new biochemical activity of Yta7 and provided in vivo evidence that supports this function. The addition of the new ChIP and co-IP experiment provide important additions to the manuscript. The demonstration that other histone chaperones or chromatin remodeling complexes do not show the same histone removal activity is also an important addition. Overall, the revised manuscript provides an important addition to the factors that act at replication forks.

Specific point:

1. For the new co-IP experiments, it is important that the authors demonstrate that they are not DNA mediated by including nuclease or EtBr in the analysis.

REVIEWER COMMENTS

Again, we would like to thank the reviewers for their time to look at the revised manuscript and are pleased that the overall response is very positive. We addressed all the remaining concerns. Below, please find the detailed responses to the reviewers' comments:

Reviewer #1 (Remarks to the Author):

The authors have done a better job at more accurately describing their work in the context of the literature now.

Other problems remain - I am underwhelmed by the new experiments to support a direct role in replication in cells, i.e. the new ChIP and IP analyses.

They show 1 primer pair for a bunch of different origins to which it binds to all of them. However, they have to show some specificity of the binding by including primer pairs for regions that it doesn't bind to, as it could just be binding all over to the chromatin. Also a temporal analysis showing it binds specifically in S phase would have been useful. Once a rigorous ChIP analysis is performed, it needs to be placed in the main figure not the supplemental.

We thank the reviewer for pointing that out. In the original ChIP data, we used 6 different primer pairs for all 6 origins, but we just included one data point as the untagged control sample - we apologize for this oversight. We now include new data comparing recruitment of Yta7 in G1 and early S phase (always plus untagged controls). Yta7 recruitment is significantly stimulated during early S phase compared to G1. This dynamic behavior clearly demonstrates specificity of our ChIP assay. Demonstrating that Yta7 does not bind to other random locations would not add more information. The new data also nicely fit with *in vivo* data, showing that origins of replication are hyper-acetylated during early S phase, suggesting a way to recruit Yta7 (Unnikrishnan et al., NSMB 2010).

As suggested, we placed the "early origin" data in the main Figure (new Fig. 1g) and we left the "late origin" data in the Supplemental (new Supplemental Fig. 2).

The IP shows that FLAG tagged Yta7 brings down some replication proteins. However, given that Yta7 binds to chromatin, they need to either add a nuclease or EtBr to show that the interaction is not through the DNA, as otherwise they are just pulling down chromatin and every protein on chromatin. Again, doing this analysis in a temporal manner would have been useful, and they must also blot for chromatin proteins that it does not bind too, to establish specificity. Once rigorous IPs are completed, they should be included in the main figure not the supplemental.

We again agree with the reviewer. We repeated the Co-IP experiment and added benzonase to eliminate DNA mediated interactions. As can be seen in new Fig. 1 f, adding benzonase does not prevent interaction of Yta7 with the replicative helicase MCM. However, it does inhibit interaction with the origin recognition complex ORC. This shows that i) Yta7 physically interacts with the replisome via MCM ii) that our previous observed ORC interaction was likely mediated by DNA and that iii) our Co-IP experiments are specific. Yta7 might also interact with soluble, non-loaded MCM

outside S phase. We feel at this point, temporal analyses of Yta7/MCM interaction would not add to the story. As requested, we included a chromatin binding protein as a negative control. **New Fig. 1f** shows that Yta7 does not physically interact with Ioc3, a subunit of the chromatin remodelling complex ISW1a, also involved in chromatin replication (Kurat et al., 2017). Together, all these experiments clearly show that our Yta7/MCM interaction is specific.

Regardless, even if they could show that Yta7 goes to replication fork, it seems more likely to be involved in events after DNA replication given that it only is recruited to acetylated chromatin and the literature on the human homolog.

They also cite the fact that human counterpart of Yta7, ATAD2, binds to replication sites in support of Yta7 doing the same. But ATAD2 specifically binds to the histone modifications that are found on newly-synthesized histones (H4K5 and 12) (Kool et al nature communications 2016) and on H3 (Revenko 2010), which would position it on the newly-replicated chromatin, not for a role in disassembling chromatin. This paper also shows that ATAD2 depletion causes a replication defect which is what you see when you get rid of factors that assemble new histones into chromatin. i.e. Stillman and others have shown that depletion of proteins that assemble newly-synthesized H3/H4 i.e. CAF-1 feedback to block replication, which could be what is happening here with ATAD2, and potentially also Yta7.

The fact that Yta7 only binds to acetylated chromatin that have H3 N-terminal acetylations (same as the ones that ATAD2 binds to) indicates that it is likely doing the same thing as ATAD2, i.e. after replication *in vivo*. The biochemical disassembly assay is likely to be an *in vitro* artifact. As such there is no evidence that Yta7 is disassembling chromatin *in vivo*.

Here we must disagree with the reviewer. In contrast what the reviewer was saying, there is a large body of papers, all *in vivo* studies and all suggesting that Yta7's mode of action is to disassemble or evict nucleosomes (Kurat et al., *Genes & Development* 2011; Lombardi *et al.*, *PNAS* 2011; Lombardi *et al.*, *Genetics*, 2014; Qui et al., *Genome Research* 2016). Importantly, this could never been confirmed *in vitro*. We now show this in detail and, uncovered a unique mechanism regulating this process. This would not have been possible without sophisticated *in vitro* biochemistry.

However, we do agree with the reviewer that the fact that ATAD2, the human Yta7, can interact with acetylation marks found on newly-synthesized histones is interesting – and perfectly in line with our proposed model in our paper. We demonstrated that Yta7 disassembles nucleosomes in a cell cycle-dependent manner in front of the replisomes to facilitate DNA synthesis. We also proposed that Yta7 might be involved in the assembly of chromatin behind the replication fork. This might happen either alone or with help of classical histone chaperones (like CAF-1 or FACT). There is growing evidence that chromatin factors, capable of destabilizing nucleosome, like FACT, are also involved in the assembly of new chromatin behind the replisome – we believe that Yta7 might act similarly. The ATAD2/histone interaction data support this model, however, with ATAD2, this chromatin assembly activity could not been confirmed *in vitro* (Koo et al., *Oncotarget*, 2016). Answering these questions is clearly an interesting goal in the future. However, we agree with the reviewer that this aspect of Yta7 biology is important and we extended the discussion.

Reviewer #2 (Remarks to the Author):

The authors have satisfactorily addressed the issues raised by this reviewer. The modifications to the text and inclusion of the additional data have strengthened the manuscript significantly.

One very minor point. As far as I understand it, *Drosophila* does not have an ATAD2/Yta7 homolog so perhaps the text in the final paragraph page 4 should read, "Yta7 is conserved among eukaryotes....." rather than "conserved among all eukaryotes".

We wish to thank the reviewer again for her/his very constructive comments on the manuscript. As requested, we changed that paragraph.

Reviewer #3 (Remarks to the Author):

The revised manuscript by Kurat and colleagues addresses the role of the Yta7 ATPase in chromosomal replication. They have identified an intriguing new biochemical activity of Yta7 and provided in vivo evidence that supports this function. The addition of the new ChIP and co-IP experiment provide important additions to the manuscript. The demonstration that other histone chaperones or chromatin remodeling complexes do not show the same histone removal activity is also an important addition. Overall, the revised manuscript provides an important addition to the factors that act at replication forks.

We are very happy that the reviewer finds that we "*have identified an intriguing new biochemical activity of Yta7 and provided in vivo evidence that supports this function*".

Specific point:

1. For the new co-IP experiments, it is important that the authors demonstrate that they are not DNA mediated by including nuclease or EtBr in the analysis.

We agree that the nuclease control is important. We repeated the Co-IP experiment and added benzonase to eliminate DNA mediated interactions. As can be seen in new Fig. 1 f, adding benzonase does not prevent interaction of Yta7 with the replicative helicase MCM. However, it does inhibit interaction with the origin recognition complex ORC. This shows that i) Yta7 physically interacts with the replisome via MCM, ii) that our previous observed Yta7/ORC interaction was likely mediated by DNA and that iii) our Co-IP experiments are specific.

REVIEWERS' COMMENTS

Reviewer #1 (Remarks to the Author):

I am satisfied with the reviewers response and additional experiments that were added. The manuscript is now rigorous.

REVIEWER COMMENTS

Again, we would like to thank the reviewers for their time to look at the revised manuscript. We are pleased that the reviewer finds the manuscript now suitable for publication

Reviewer #1 (Remarks to the Author):

I am satisfied with the reviewers response and additional experiments that were added. The manuscript is now rigorous.